# Gene regulatory networks involved in activation of Notch signaling by AGEs in the pathogenesis of diabetic kidney disease

Somorita Baishya[1,2], Adyasha Sarangi[1], Pramod R. Somvanshi[3]*,
Anil Kumar Pasupulati[1,4]*

1 Department of Biochemistry, University of Hyderabad, Hyderabad, Telangana, India, 2 Department of Biotechnology, School of Sciences, Woxsen University, Hyderabad, Telangana, India, 3 Department of Systems and Computational Biology, University of Hyderabad, Hyderabad, Telangana, India, 4 Division of Renal Medicine, Department of Medicine, Brigham and Women's Hospital, Harvard Medical School, Boston, Massachusetts, United States of America

* pramodrs@uohyd.ac.in (PRS); anilkumar@uohyd.ac.in (AKP)

## Abstract

Diabetic kidney disease (DKD) is a progressive disease characterized by early events such as podocyte injury followed by glomerulosclerosis, varying degrees of protein-uria, decreased glomerular filtration rate, and eventual organ failure. Podocytes, essential for glomerular permselectivity, are targets for an array of noxious stimuli in diabetes. Notch signaling, critical for podocyte differentiation during nephrogenesis, reactivates in mature podocytes, as evidenced in DKD patients and animal models. Notch reactivation in adult podocytes implicated in de-differentiation and apoptosis. Although elevated advanced glycation end-products (AGEs), heterogeneous mole-cules derived from non-enzymatic glycation, were paralleled with Notch reactivation in podocytes, the precise mechanism remains unknown. This study identified a condensed network of 58 genes modulated by AGEs. These genes non-canonically reactivate Notch by suppressing the PI3K-AKT pathway and activating NF-kB sig-naling, affecting podocyte biology and function. The study provides novel information about reno-toxic events and new therapeutic targets to prevent podocyte injury and kidney dysfunction.

## Introduction

According to WHO, approximately 422 million individuals globally have diabetes, with the majority residing in low- and middle-income nations. The disease is directly responsible for 1.5 million fatalities annually (WHO, 2024). Over the past few decades, there has been a steady rise in both the number of cases and the inci-dence of diabetes [1]. Persistent hyperglycemia is a hallmark of diabetes mellitus (DM), which can be caused by insulin resistance (type II) or an absolute or relative insulin shortage (type I) [2]. DM is one of the leading causes of the development of

**Data availability statement:** The datasets analyzed in the current study are available in the GEO database (https://www.ncbi.nlm.nih.gov/geo/query/acc.cgi?acc=GSE30122). The accession number (GSE30122) is provided in the article. All data generated and analyzed during this study are included in the article and its supplementary information files.

**Funding:** The authors acknowledge the support of Rajiv Gandhi Centre for Biotechnology, Department of Biotechnology, Ministry of Science and Technology, India DBT-RA/2023/January/NE/3723 Dr Somorita Baishya.

**Competing interests:** The authors have declared that no competing interests exist.

kidney problems; around 30% of people with type I and 10% to 40% of those with type II DM will eventually experience renal failure [3]. The longer a patient suffers from DM, the greater their risk of developing secondary complications of diabetes, including diabetic kidney disease (DKD). DKD is widespread across the globe, adversely affecting human health with substantial economic burdens on human society [4]. DKD is a significant contributor to chronic kidney disease (CKD) [5]. DKD is presented with functional alterations such as decreased glomerular filtration rate (GFR) and varying degrees of proteinuria [6]. It also leads to altered metabolic, inflammatory, haemodynamic signalling pathways along with several molecular mediators that creates a feedback loop in promoting general kidney damage [7]. Structural alterations that prevail in DKD include glomerulosclerosis, thickening of the glomerular basement membrane, and podocytopathy [8]. Podocytes, the visceral cells of the glomerulus, provide epithelial coverage to the capillaries, thereby playing an instrumental role in the permselectivity and preventing protein loss into the urine. Podocytes are terminally differentiated cells with very limited proliferative potential, therefore an absolute podocyte number and integrity are critical factors in maintaining normal renal function [6].

Podocytes are targets of intensive insults in various clinical conditions [9]. Podocyte injury is considered an early insult in the settings of DKD, and injury of podocytes parallels with proteinuria [10]. For instance, the transition from epithelial to mesenchymal phenotype is one of the insults to podocytes in the settings of DKD [11,12]. This phenomenon of de-differentiation of podocytes is an outcome of the reactivation of Notch signaling in the adult podocytes [13]. Notch signaling is crucial for fate determination during embryological development; particularly, it is warranted for the differentiation of progenitors during nephrogenesis [12]. This juxta cellular signaling is activated when one cell's ligand (Delta or Jagged) interacts with the adjacent cell's Notch receptor (Notch 1–4). Notch ligand and receptor interaction triggers proteolytic cleavage of the Notch intracellular domain (NICD), which could trigger the expression of target genes upon translocation to the nucleus. [11]. However, in terminally differentiated podocytes Notch signaling is generally minimal and inactive [14]. Accumulated evidence suggests that reactivation of Notch signaling leads to the loss of terminal differentiation markers and forces podocytes to re-enter the cell cycle, thereby creating deleterious effects [12,14–16] while prevention of Notch reactivation showed nephroprotective effects [11,17,18].

Chronic hyperglycemia leads to the accumulation of advanced glycation end-products (AGEs) in various tissues, contributing to microvascular complications in the eye lens, retina, and kidney, as well as macrovascular complications in the arteries of the heart, brain, and lower limbs. These pathogenic processes are strongly implicated in the development of cataracts, diabetic retinopathy, diabetic kidney disease (DKD), cardiovascular disease, cerebrovascular disease, and peripheral vascular disease [19–23]

Several noxious stimuli that prevail in diabetes settings were shown to induce reactivation of Notch signaling in adult podocytes. They include glucose, growth hormone, and advanced glycation end-products (AGEs) [11,24–26]. During chronic

hyperglycemic conditions, glucose reacts with a free-amino group of amino acids and eventually forms glycation adducts on client proteins by a series of non-enzymatic reactions [27,28]. Such chronic exposure leads to the accumulation of AGEs in various tissues, contributing to microvascular complications in the eye lens, retina, and kidney as well as macrovascular complications in the arteries of the heart, brain, and lower limbs. These processes are implicated in the pathogenesis of several complications, including cataracts, retinopathy, DKD, cardiovascular disease, cerebrovascular disease, and peripheral vascular diseases [29–31]. AGEs also interact with lipids and nucleic acids, elicit cellular and molecular changes, and orchestrate the end-organ pathology in diabetes and aging settings. Kidneys eliminate AGEs circulating in the bloodstream through glomerular filtration. However, elevated levels of AGEs due to hyperglycemia result in insufficient removal, increasing their concentration. Excess AGEs in the circulation or accumulated AGEs in the kidney increases the risk of renal dysfunction [11,32–37]. Upon binding to their receptors (RAGEs), AGEs trigger anomalous signaling cascades [38]. Recent study have also found upregulation of RAGE expression in SGLT2-mediated glucose uptake in proximal tubular cells, being involved in the tubulointerstitial [39]. However, the precise mechanism of Notch activation by these cytotoxic AGE molecules is largely unknown.

Generally, for eliciting a response, proteins work in bio-molecular networks utilizing multiple feedback loops to accomplish a signal transduction. The topology and kinetics of the network dictate its output response [40,41]. Hence, we took a holistic approach to understanding the Notch reactivation mechanism. We used dataset generated by high throughput technologies to get insights into the molecular and cellular players in the reactivation. This panoramic view generated a condensed network of 58 genes that predicted that AGEs fostered by chronic hyperglycemia suppress PI3K-AKT pathway, activate NF-kB signaling and elicit inflammatory responses which in turn reactivates Notch signaling Reactivation of Notch also suppresses PI3K-AKT signaling. This induces apoptosis in the terminally differentiated podocytes. These findings can be exploited to develop treatments that can stop podocyte injury and can cure Diabetic nephropathy (DN).

## Materials and methods

### Data acquisition

The gene expression profile for microarray data was collected from Gene ExpressionOmnibus (GEO) database through the keywords "DKD" or "DN". We downloaded and analyzed the expression profile of GSE30122 from GPL571 platform, that had 19 DN and 50 Control. GSE30122 contained both glomerular and tubular specimens. Raw data for the dataset was contained in zipped. TAR files were downloaded and untarred manually to obtain individual.CEL files. "GEOquery" (v2.70.0) [42] package was used for accessing GEO SOFT files for each dataset..CEL files for the dataset was parsed to R studio using "ReadAffy" of "Affy" (v 1.80.0) package [43]. To confirm homogeneity across datasets and evaluation of their quality we performed background correcting, data normalization, batch effect removal using "rma" from "Affy" (v 1.80.0) package and "ComBat" from "sva" package [44]. Subsequently, probe annotation was performed and probes with multiple genes were excluded. Genes mapping to multiple probes were calculated as average.

### Identification of differentially expressed genes

Differentially expressed genes (DEGs) from the dataset were identified using "limma" package (v 3.58.1) of R software with $|log2FC|>1$ and $p<0.001$ considered significant [45]. Volcano plot for the DEGs were visualized using "ggplot2" package (v 3.4.4).

### Functional enrichment analysis

To understand the significant pathways involved in DN and to identify the genes involved with Notch signaling and AGE-RAGE signaling we performed their Gene Ontology (GO) and Kyoto Encyclopedia of Genes and Genomes (KEGG) pathway analysis of the DEGs. The GO resource tool provides annotation for molecular function (MF), cellular component

(CC), and biological process (BP) of genes to facilitate the functional aspects of genes during computational analysis. KEGG analysis was performed to assess the functional aspect of the genes. "Clusterprofiler" was employed to give insights into the enrichment analysis of DEGs [46]. R package "org.Hs.e.g.,db" was used for conversion between gene IDs. Cluepedia app of Cytoscape was used to generate and explore the expression, activation, and inhibitory relationship between the gene interaction network [47,48]. STRING Action File in Cluepedia panel was used as source for action query and the interactions were set to different colours to indicate the action. The action were based on a statistical method called Kappa scoring, which range from 0 to 1, but can also be customized.

## Gene set enrichment analysis

Gene set enrichment analysis (GSEA) was performed using "clusterProfiler" (v 4.10.0) to determine the pathways that were enriched in the dataset [46]. This analysis uses set of predefined genes to determine whether the pooled genes represent specific well-defined biological states and display coherent expression. The top "HALLMARK" pathway terms were depicted based on Net Enrichment Score (NES), p, and gene ratio. P was set at 0.05, minimum and maximum gene set size was set at 25 and 500 respectively and Benjamini–Hochberg (BH) method was applied to adjust the p for the false discovery rate (FDR) in our study. The tests were regarded as significant with an adjusted p threshold of 0.05 and gene count ≥2. The gene set database was downloaded from Human MSigDB Collections.

## PPI network creation and identification of subnetworks

Protein-protein interaction (PPI) network for the genes involved with the top 5 pathways along with Notch signaling and AGE-RAGE signaling was generated using STRING app integrated with Cytoscape 3.10.1. confidence level 0.90 [48,49]. The highly interconnected subnetworks (clusters/ hubs) of the merged network were identified using MCODE plugin of Cytoscape software, where degree cutoff = 2, node score cutoff = 0.2, and k-core = 2 was considered as filtering criteria [50].

## Analysis of topological parameters

Network Analyzer plugin of Cytoscape was utilized to check the central measures of the nodes [51]. Among the various central measures provided by Network Analyzer, we based our study on the degree, betweenness centrality (BC) and clustering coefficient. Degree of a node indicates the number of node connections a particular node has, whereas BC indicates the frequency with which a node is located on the shortest path linking other nodes and clustering coefficient reflects the strength of degree. We discarded nodes having clustering co-efficient > 0.5 in order to focus on a tight network. Thus, analysing topological properties will give insights into the functioning and essentiality of the nodes at the systemic level [52]. To understand more deeply correlation between the genes a correlation plot was generated using "corplot" of R studio.

## Validation with RNA-Seq data

To independently validate the transcriptional changes identified from the microarray dataset GSE30122, we performed a complementary analysis of publicly available RNA-seq data (GSE299230), which profiles gene expression in human kidney tissue under DN–relevant conditions. Raw FPKM values were retrieved from GEO and processed in R studio. Genes with FPKM > 1 in at least two samples were retained to remove low-abundance transcripts, and expression values were log2-transformed.

Differential expression analysis was performed using the limma-voom pipeline, with statistical thresholds set at adjusted p-value < 0.01 (Benjamini–Hochberg correction) and absolute log2 fold-change > 0. Overlapping genes between RNA-seq DEGs and the 167 AGE-associated genes from the microarray dataset were identified using base R functions.

Venn diagram of the genes common between the 2 platforms was constructed using https://bioinformatics.psb.ugent.be/webtools/Venn/.

## Validation of existing dataset with regional transcriptomic datasets

The significant driver genes identified based on centrality and statistical measures were cross-referenced to the publicaly available Kidney Precision Medicine Project (KPMP) database (https://atlas.kpmp.org/). We checked the regional transcriptomic expression of the key genes to find consistency and relevance of our findings with regards to AGE-induced DN. A violin plot was plotted to compare the logFC distribution of the expressions from the existing and the independent dataset and scatter plot with connected lines was plotted to using "ggplot2" package (v 3.4.4).

## Statistical analysis

All statistical analysis were performed on R software (version 4.3.2).

## Results

### 1620 differentially expressed genes (DEGs) identified between control and DN patients

Fig 1 depicts the workflow of the present study. Post pre-processing and annotating GSE30122 we performed differential gene expression analysis with |log2FC|>0 and p<0.01. We identified 1620 genes were differentially expressed between healthy (control) and DN patients, which included 960 upregulated and 660 downregulated (Supplementary table 1 in S1 File). The DEGs are represented as a volcano plot in Fig 2.

### PI3K-Akt signaling pathway, inflammatory pathways and epithelial mesenchymal transition processes are most enriched in DN condition

We tried to dive into the pathways that are most enriched in diabetic condition and identify genes associated with Notch signaling and AGE-RAGE signaling (Supplementary table 2 in S1 File). KEGG pathway analysis revealed PI3K-AKT pathway is the most enriched pathway in DN condition. 55 genes were involved in this pathway. Apart from this Cytokine-cytokine receptor interaction, MAPK signaling pathway, Focal adhesion, and Regulation of actin cytoskeleton pathways were the top 5 pathways that were enriched in DN condition (Fig 3A, Table 1). We also found 21 genes were involved in AGE-RAGE signaling and 3 genes were involved with Notch signaling (Supplementary table 2 in S1 File). We also studied the hallmarks enriched across the dataset using GSEA and found hallmarks for COMPLEMENT, INTERFERON GAMMA RESPONSE, and EPITHELIAL MESENCHYMAL TRANSITION were the top 3 hallmarks descriptions based on NES (Fig 3B, Supplementary table 3 in S1 File). KEGG pathway analysis helped in identifying 167 unique genes with which we carried on our further analysis (Table 2).

### 58 genes formed a tightly regulated network

STRING integrated in Cytoscape was used to generate a PPI network for 167 DEGs at 0.90 confidence level. 9 highly connected regions, or clusters were found in the network, according to module analysis done with MCODE. We also studied the topological properties of the gene using Network Analyzer plugin of the software. We checked for their betweenness centrality, degree, and clustering co-efficient and a tightly regulated network with 58 genes (Table 3). CXCR4, CXCL8, and CCR5 had highest number of connections.

### Chemokine markers, ECM/ fibrotic markers and stress response markers were the central players

To get more comprehensive idea about how these 58 genes interact with each other, we used Cluepedia app from Cytoscape. We realized these genes were majorly involved with Cytokine-cytokine receptor interaction, Regulation of actin cytoskeleton, PI3K-Akt signaling pathway, Chemokine signaling pathway, Focal adhesion, and NF-kB signaling pathway. Cluepedia showed CXCL8, CCL2, CCR5, CXCR4, COL1A2, COL6A2, LAMC1, ITGB7, SYK, IRAK1, TNFRSF1B, GADD45B were the key players in the network (Fig 4A). The expression pattern of the 58 genes across samples has been

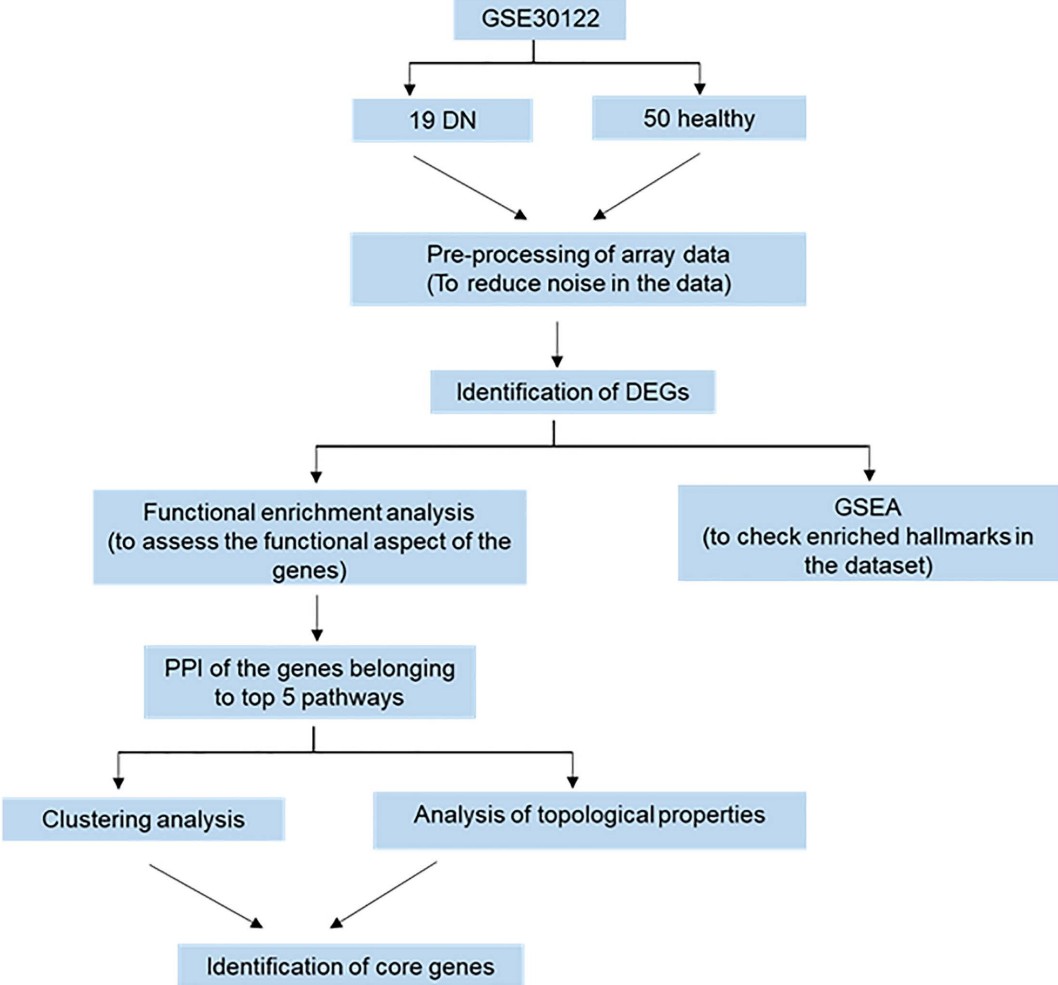

**Fig 1. Scheme of study workflow.** GSE30122 was acquired fromGEO Database. Post pre-processing data in R studio, differentially expressed genes (DEGs) were identified using "limma". Gene set enrichment analysis (GSEA) and functional enrichment analysis was performed using "ClusterProfiler" to identify the hallmarks and pathways enriched in pathogenesis of DN. PPI network was generated in Cytoscape with the genes involved in top 5 pathways at 0.90 confidence level. Clustering analysis and analysis of topological parameters were performed to identify the core gene network using MCODE and Network Analyzer respectively. Cluepedia was used for visualising the genes and their pathway association.

depicted as a heatmap in Fig 4B. Among several cytokines, CCL2 is a significant driver based on its logFC and centrality measures. To get better insights into the interaction of the pathways and the genes involved we constructed a correlation plot. The plot was constructed to highlight genes whose p values were ≤ 1e-14 and identified 27 genes, which was plotted as a matrix (Fig 4C). The plot elucidated that except for IKBKG, LAMB1, MAGI2, and PAK4 all other genes were positively correlated. These 4 genes are also downregulated in DN condition.

## 15 genes were found to be common upon cross-platform validation with RNA-Seq data

To independently test whether the presumed core genes were differentially expressed in a DKD-relevant system, we analyzed RNA-seq dataset GSE299230. From GSE299230, a total of 12665 DEGs were identified, of which 246 were upregulated and 344 were downregulated significantly. Comparison with the 167 AGE-related genes from GSE30122

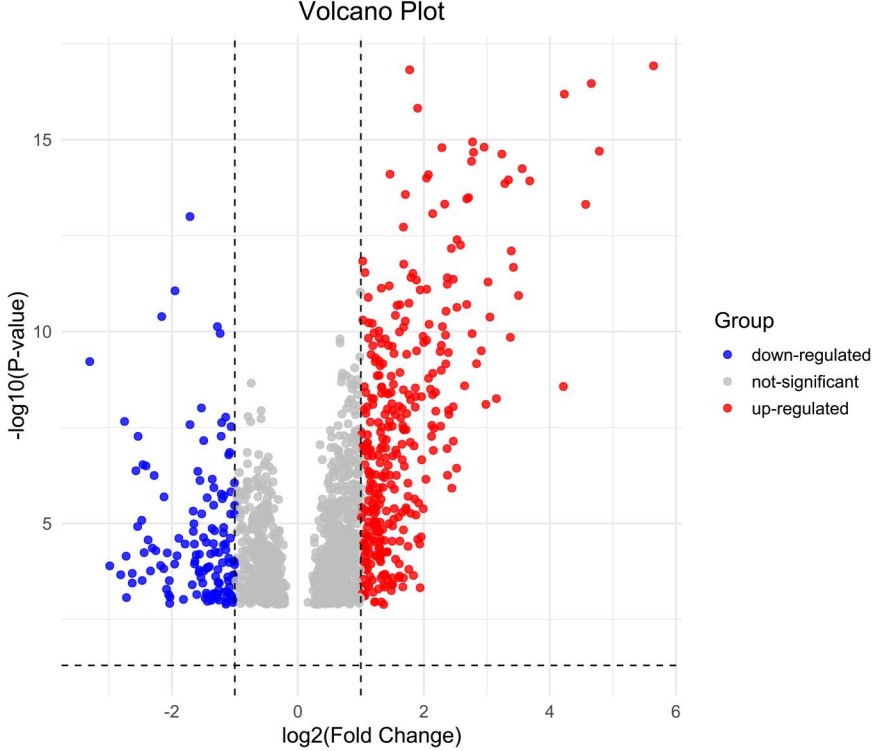

**Fig 2. 1620 DEGs identified.** DEGs from the dataset were identified using "limma" package of R software with |log2FC|>0 and p<0.01 considered significant. Volcano plot for the DEGs were visualized using "ggplot2" package. 960 genes denoted in red were upregulated and 660 gene depicted in blue were downregulated and the non-significant genes are denoted in grey (|log2FC|>0 and p<0.01).

revealed 15 common genes: CCND2, THBS2, TNFSF10, EDN1, TNC, GADD45G, CSF1R, MYL12B, MYD88, LAMC2, DDIT3, MMP2, EGF, GADD45A, and CCL2 (Fig 5A, Table 4). DDIT3, GADD45A, THBS2, CCL2, and CSF1R showed consistent expression trends across platforms (Fig 5B and 5C), and are known mediators of inflammation, extracellular matrix remodeling, and stress responses in diabetic kidney disease. DDIT3 and GADD45A were consistently downregulated, while THBS2, CCL2, and CSF1R were consistently upregulated (Fig 5D and 5E). However, DDIT3, THBS2, and CSF1R were not part of the sub-network of 58 genes on which we are focussing in this work. This partial convergence highlights both the strengths and the contextual limits of our findings. This computational replication, nevertheless, strengthens the credibility of the proposed network.

### Five genes showed consistent trends in the independent dataset validation

The existing microarray dataset was also cross-validated with independent regional transcriptomic datasets available in KPMP database (https://atlas.kpmp.org/). The violin plot (Fig 4D) showed the distribution of expressional fold changes of the 12 driver genes across the microarray dataset (existing dataset) (blue violin) and regional transcriptomic dataset (independent dataset) (red violin) from KPMP public database. Greater variability can be seen in the transcriptomic dataset than the microarray. We observed of the 12 key driver genes (CXCL8, CCL2, CCR5, CXCR4, COL1A2, COL6A2, LAMC1, ITGB7, SYK, IRAK1, TNFRSF1B, GADD45B) 5 genes viz. COL6A2, ITGB7, IRAK1, TNFRSF1B, GADD45B showed similar expression trends strengthening the relevance of these genes in the context of AGE-induced DN (Fig 4E, Supplementary table 4 in S1 File). CXCL8 is upregulated in both sets but shows stronger expression in transcriptomic dataset. Differences in the distribution projected by violin plot may be due to platform-specific biases.

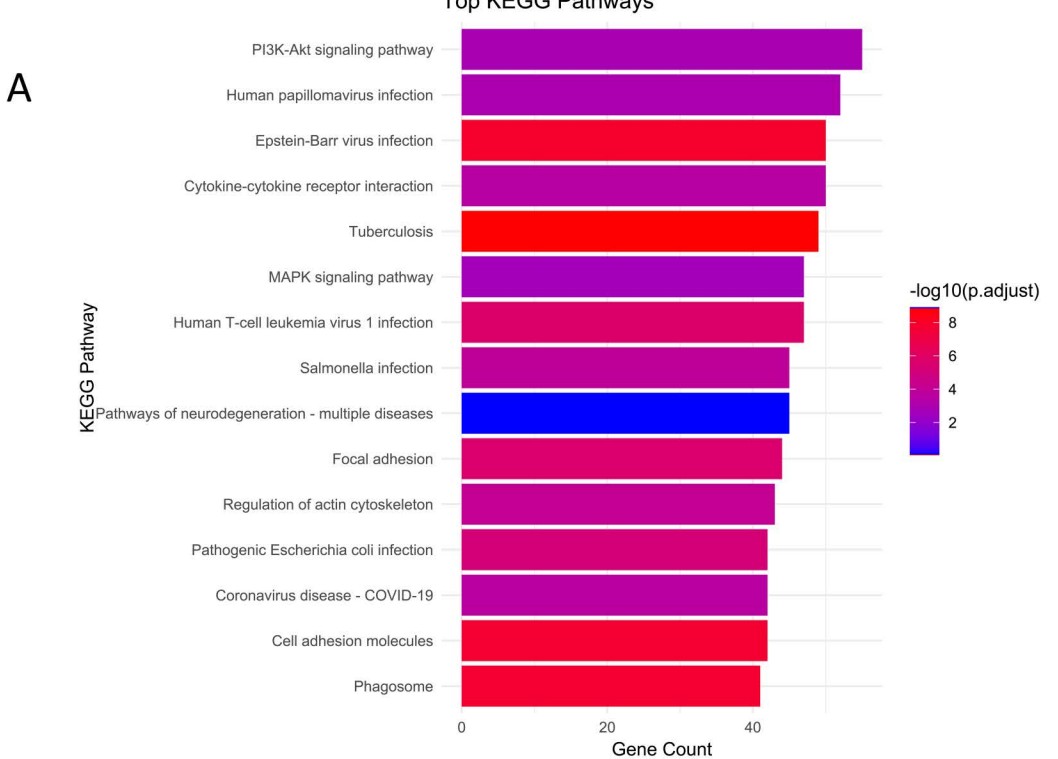

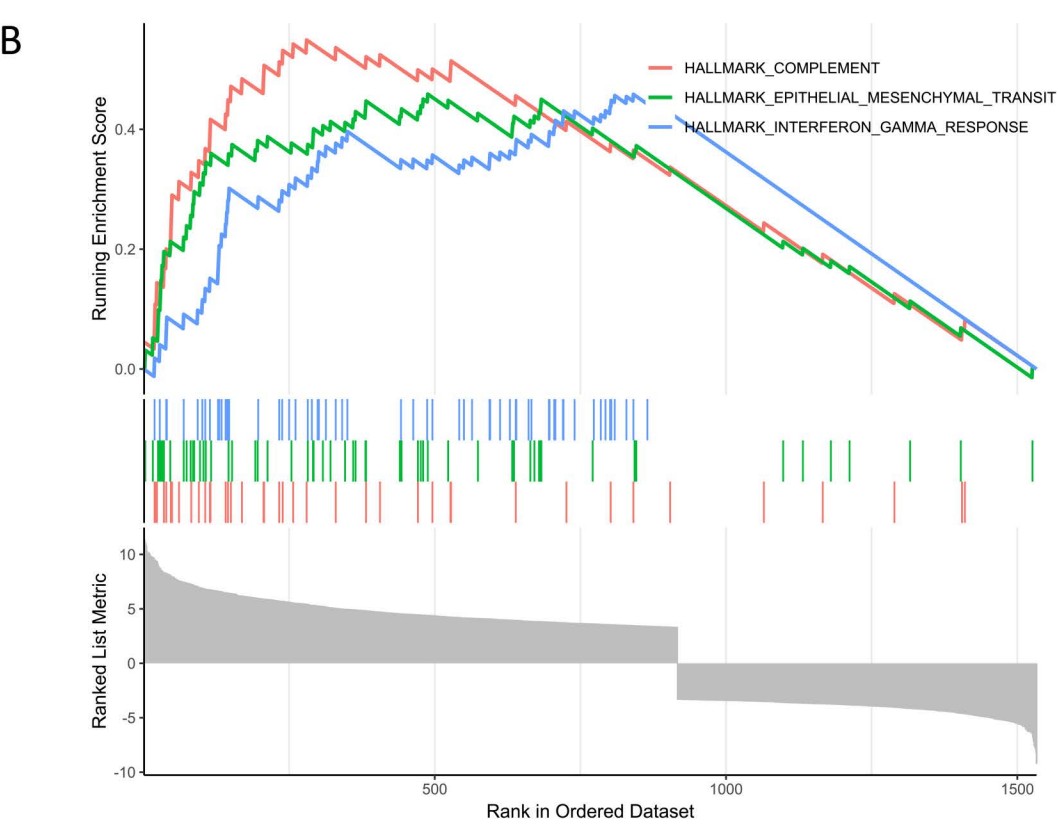

**Fig 3. PI3K-Akt signaling pathway, inflammatory pathways and epithelial mesenchymal transition processes are most enriched in DN condition.** A. KEGG pathway analysis shows the top 15 enriched pathways are represented as bar plot. PI3K-AKT pathway, Cytokine-cytokine receptor interaction, MAPK signaling pathway, focal adhesion and regulation of actin cytoskeleton are the most enriched pathways in DN condition. B. Gene set enrichment analysis (GSEA) shows the complement, interferon gamma response, and epithelial mesenchymal transition hallmarks upregulated in DN pathogenesis based on NES, meaning that inflammation and fibrosis- related pathways are upregulated in DN condition.

**Table 1. Top 5 pathways and their genes from KEGG analysis.**

| ID | Description | pvalue | p.adjust | Count | qvalue |
|---|---|---|---|---|---|
| hsa04151 Gene ID | PI3K-Akt signaling pathway | 0.000219 | 0.001299 | 55 | 0.000874 |
| | COL1A2/TNC/COL6A3/THBS2/FN1/FGF9/IL7R/CSF1R/LAMC2/TLR2/SYK/VWF/ITGA4/LPAR1/EGF/COMP/NGF/ GHR/SPP1/ITGA2/IL7/PDGFRA/YWHAH/LAMA2/PTEN/RXRA/MAGI2/VEGFA/HGF/LAMC1/CCND2/FLT1/JAK1/ MYC/PHLPP1/COL4A1/PIK3CB/ITGB7/COL9A1/EFNA4/MET/FGF7/CCNE1/LAMB3/IFNAR2/EFNA1/LAMB1/ FLT3LG/RPS6/TCL1B/IKBKG/IL2RB/CDK4/COL6A2/FGF4 | | | | |
| hsa04060 Gene ID | Cytokine-cytokine receptor interaction | 3.35E-05 | 0.000288 | 50 | 0.000194 |
| | CXCL6/IL10RA/CCR2/ACKR4/FAS/IL7R/CSF1R/CCL19/CCR7/IL33/CCL5/CCL2/LTB/IFNGR2/TNFSF15/BMP7/ CXCR4/CX3CR1/TNFRSF1B/NGF/CD27/GHR/TNFSF10/IL7/CXCL1/CCR5/CCR6/IL32/CXCR6/TNFRSF17/ PF4V1/CTF1/TNFRSF1A/XCL1/IL18/BMP5/TNFRSF11B/CXCL12/IL21R/IFNAR2/INHBA/ACVR2A/IL2RB/CD40/ CXCL8/CCL18/IFNGR1/IL16/CCL21/IL37 | | | | |
| hsa04010 Gene ID | MAPK signaling pathway | 0.000307 | 0.00176 | 47 | 0.001184 |
| | FGF9/FAS/CSF1R/MYD88/PLA2G4A/GADD45B/EGF/RAC2/PRKCB/CASP3/NGF/RASA2/MAP4K1/PDGFRA/ MAPK13/CACNB2/VEGFA/HGF/RPS6KA3/GNA12/FLT1/MYC/TNFRSF1A/GADD45G/CD14/MAPK8/HSPA1L/ EFNA4/IRAK1/NLK/DUSP1/MET/FGF7/RPS6KA1/CACNA2D3/GADD45A/MAPK10/ELK1/EFNA1/FLT3LG/PAK2/ IKBKG/CDC25B/DDIT3/MAPK8IP2/CACNA1D/FGF4 | | | | |
| hsa04510 Gene ID | Focal adhesion | 8.90E-08 | 2.43E-06 | 44 | 1.63E-06 |
| | COL1A2/TNC/COL6A3/THBS2/FN1/ACTN1/LAMC2/VWF/ITGA4/EGF/COMP/RAC2/PRKCB/BIRC3/SPP1/ITGA2/ PDGFRA/PIP5K1C/LAMA2/PTEN/ACTB/VAV1/VEGFA/HGF/LAMC1/CCND2/FLT1/COL4A1/PIK3CB/ITGB7/ COL9A1/MAPK8/MET/LAMB3/MAPK10/ELK1/PAK6/LAMB1/MYL10/PAK2/PAK4/MYL12B/COL6A2/CTNNB1 | | | | |
| hsa04810 Gene ID | Regulation of actin cytoskeleton | 8.04E-06 | 8.48E-05 | 43 | 5.70E-05 |
| | ITGB2/FN1/FGF9/ACTN1/ITGAM/BAIAP2/C7/PFN1/ENAH/ARPC1B/ITGA4/LPAR1/MSN/EGF/RAC2/CXCR4/ GNA13/ITGA2/PDGFRA/PIP5K1C/ACTB/VAV1/GNA12/PIK3CB/ITGB7/FGF7/SSH1/ITGAX/ITGAL/ARPC3/ CXCL12/TMSB4Y/PAK6/MYL10/PAK2/ACTR3/PAK4/MYL12B/SCIN/RDX/KNG1/C6/FGF4 | | | | |

# Discussion

Persistent hyperglycemia results in the accumulation of AGEs in both tissues and blood. Kidneys play major role in clearing out the AGEs, hence their accumulation has been linked to the development of diabetic nephropathy, which is a major complication of diabetes and increases risk for end-stage renal disease and mortality [32,53]. Recent study from our laboratory suggests that elevated levels of AGEs are implicated in the reactivation of Notch signaling in podocytes. Notch signaling is crucial for fate determination of podocytes during embryological development, however, reactivation of Notch in adult podocytes induces loss of epithelial features and compromises its permselective potential [11]. However, the mechanistic understanding of how AGEs induce EMT is still lacking. Uncovering the signaling events that integrate and activate Notch signaling during exposure to AGEs could potentially improve our understanding about the pathophysiology of diabetic kidney disease and also guide us to develop novel intervention strategies. Network biology aids in understanding large-scale characteristics of various biological processes and in identifying specific biological attributes [54]. Therefore, we utilized network biology approach to identify genes and pathways that might play crucial role in reactivating Notch signaling.

We performed an array of bioinformatic analyses on GSE30122 to draw a gene interaction network that can highlight the molecular pathways that implicate the reactivation of Notch signaling in AGEs induced DN. Functional enrichment

**Table 2. 167 genes identified from KEGG pathway analysis that are closely associated with DN pathogenesis along with their module and topological properties.**

| SYMBOL | log2FC | p | Cluster | Clustering co-efficient | Degree | Betweeness Centrality |
|---|---|---|---|---|---|---|
| ACKR4 | 1.18 | 6.12E-11 | Unclustered | 0.833333 | 4 | 1.06E-05 |
| ACTB | 0.78 | 4.87E-06 | Cluster 3 | 0.066667 | 15 | 0.161055 |
| ACTN1 | 1.44 | 2.30E-10 | Unclustered | 1 | 2 | 0 |
| ACTR3 | 0.79 | 4.87E-04 | Cluster 3 | 1 | 3 | 0 |
| ACVR2A | 0.52 | 5.18E-04 | Unclustered | 0 | 3 | 1 |
| ARPC1B | 1.67 | 2.81E-08 | Cluster 3 | 1 | 3 | 0 |
| ARPC3 | 1.38 | 2.30E-04 | Cluster 3 | 1 | 3 | 0 |
| BAIAP2 | −0.74 | 2.21E-09 | Unclustered | 0.333333 | 3 | 0.006713 |
| BIRC3 | 1.44 | 3.05E-07 | Cluster 4 | 0.4 | 6 | 0.052912 |
| BMP5 | 0.47 | 1.20E-04 | Unclustered | 0 | 1 | 0 |
| BMP7 | −1.08 | 1.41E-07 | Unclustered | 0 | 1 | 0 |
| C6 | 0.76 | 0.001029 | Unclustered | 0 | 1 | 0 |
| C7 | 2.65 | 2.59E-09 | Unclustered | 0 | 1 | 0 |
| CACNA1D | 0.46 | 0.001226 | Unclustered | 0 | 2 | 0.031492 |
| CACNA2D3 | 0.57 | 1.91E-04 | Unclustered | 0 | 1 | 0 |
| CACNB2 | −0.91 | 1.27E-05 | Unclustered | 0 | 2 | 0.015873 |
| CASP3 | 0.97 | 2.07E-07 | Unclustered | 0 | 7 | 0.140125 |
| CCL18 | 0.51 | 8.92E-04 | Unclustered | 0 | 1 | 0 |
| CCL19 | 2.84 | 6.86E-10 | Cluster 1 | 0.5 | 12 | 0.006027 |
| CCL2 | 1.86 | 9.33E-09 | Cluster 1 | 0.6 | 10 | 0.028701 |
| CCL21 | 0.64 | 0.001182 | Cluster 1 | 0.575758 | 12 | 0.004554 |
| CCL5 | 2.47 | 9.09E-09 | Cluster 1 | 0.45 | 16 | 0.012644 |
| CCND2 | 0.80 | 2.56E-05 | Unclustered | 0 | 1 | 0 |
| CCNE1 | 0.48 | 1.40E-04 | Unclustered | 1 | 2 | 0 |
| CCR2 | 1.45 | 6.43E-12 | Cluster 1 | 0.781818 | 11 | 6.79E-04 |
| CCR5 | 0.83 | 1.25E-05 | Cluster 1 | 0.564103 | 13 | 0.019444 |
| CCR6 | 0.48 | 1.45E-05 | Unclustered | 0.761905 | 7 | 2.56E-04 |
| CCR7 | 1.06 | 2.74E-09 | Cluster 1 | 0.487179 | 13 | 0.044848 |
| CD14 | 1.31 | 7.38E-05 | Unclustered | 0 | 1 | 0 |
| CD27 | 1.14 | 5.49E-07 | Unclustered | 0 | 2 | 0.018803 |
| CD40 | 0.31 | 6.30E-04 | Cluster 4 | 0.4 | 6 | 0.036619 |
| CDC25B | 0.75 | 5.42E-04 | Unclustered | 0 | 2 | 0.015873 |
| CDK4 | 0.49 | 6.21E-04 | Unclustered | 0.333333 | 3 | 0.015873 |
| COL18A1 | 3.34 | | Unclustered | 0 | 1 | 0 |
| COL1A2 | 2.52 | 1.12E-14 | Unclustered | 0.5 | 5 | 0.023492 |
| COL3A1 | 1.42 | 3.64E-07 | Unclustered | 0.666667 | 4 | 0.00781 |
| COL4A1 | 0.48 | 5.46E-05 | Unclustered | 0 | 0 | 0 |
| COL6A2 | 3.39 | 8.19E-04 | Unclustered | 1 | 2 | 0 |
| COL6A3 | −0.43 | 7.94E-13 | Unclustered | 0.666667 | 3 | 6.35E-05 |
| COL9A1 | 1.35 | 7.10E-05 | Unclustered | 0 | 0 | 0 |
| COMP | 1.35 | 6.10E-08 | Unclustered | 0 | 1 | 0 |
| CSF1R | 0.68 | 6.83E-10 | Unclustered | 0 | 1 | 0 |
| CTBP1 | −0.67 | 5.00E-04 | Unclustered | 0 | 1 | 0 |
| CTF1 | 0.89 | 3.34E-05 | Unclustered | 0 | 0 | 0 |
| CTNNB1 | 1.97 | 8.41E-04 | Unclustered | 0.133333 | 6 | 0.058742 |

*(Continued)*

**Table 2.** (Continued)

| SYMBOL | log2FC | p | Cluster | Clustering co-efficient | Degree | Betweeness Centrality |
|---|---|---|---|---|---|---|
| CX3CR1 | 1.60 | 2.19E-07 | Unclustered | 0.666667 | 7 | 3.38E-04 |
| CXCL1 | 1.14 | 6.72E-06 | Cluster 1 | 0.6 | 10 | 0.015026 |
| CXCL12 | 4.66 | 2.38E-04 | Cluster 1 | 0.45 | 16 | 0.136083 |
| CXCL6 | 0.98 | 3.44E-17 | Unclustered | 1 | 3 | 0 |
| CXCL8 | 1.39 | 8.33E-04 | Cluster 1 | 0.538462 | 13 | 0.027678 |
| CXCR4 | 0.52 | 1.92E-07 | Cluster 1 | 0.5 | 13 | 0.097876 |
| CXCR6 | 0.71 | 1.97E-05 | Unclustered | 0.833333 | 4 | 1.06E-05 |
| CYBB | −0.72 | 1.21E-09 | Unclustered | 0 | 3 | 0.026234 |
| DDIT3 | −1.00 | 5.57E-04 | Unclustered | 0 | 1 | 0 |
| DUSP1 | 0.82 | 1.18E-04 | Cluster 6 | 1 | 2 | 0 |
| DUSP16 | −0.71 | | Unclustered | 0 | 1 | 0 |
| EDN1 | 0.70 | 1.98E-04 | Unclustered | 0 | 1 | 0 |
| EFNA1 | −2.54 | 3.83E-04 | Unclustered | 0 | 0 | 0 |
| EFNA4 | −0.44 | 9.19E-05 | Unclustered | 0 | 0 | 0 |
| EGF | 1.08 | 5.37E-08 | Unclustered | 0.178571 | 8 | 0.044319 |
| ELK1 | 1.12 | 3.28E-04 | Cluster 6 | 0.333333 | 3 | 0.006117 |
| ENAH | −0.29 | 3.88E-09 | Unclustered | 0.666667 | 3 | 3.34E-04 |
| FAS | 0.44 | 1.49E-10 | Unclustered | 0.333333 | 3 | 0.022936 |
| FGF4 | −1.95 | 0.001278 | Unclustered | 0 | 1 | 0 |
| FGF7 | −0.77 | 1.32E-04 | Unclustered | 0.666667 | 3 | 8.30E-04 |
| FGF9 | 0.27 | 8.68E-12 | Unclustered | 0 | 1 | 0 |
| FLT1 | 1.80 | 2.95E-05 | Unclustered | 0.166667 | 4 | 0.003144 |
| FLT3LG | −1.31 | 4.07E-04 | Unclustered | 0 | 2 | 1 |
| FN1 | −1.53 | 3.89E-12 | Cluster 2 | 0.153846 | 14 | 0.169605 |
| GADD45A | −1.56 | 2.40E-04 | Cluster 7 | 1 | 2 | 0 |
| GADD45B | −2.28 | 9.80E-09 | Cluster 7 | 1 | 2 | 0 |
| GADD45G | −0.81 | 6.35E-05 | Cluster 7 | 1 | 2 | 0 |
| GHR | −0.54 | 5.63E-07 | Unclustered | 0 | 0 | 0 |
| GNA12 | −0.95 | 2.53E-05 | Cluster 9 | 0.333333 | 4 | 0.023683 |
| GNA13 | 0.58 | 3.03E-07 | Cluster 9 | 0.666667 | 3 | 0.00781 |
| HEY1 | −0.50 | 2.10E-04 | Unclustered | 0 | 0 | 0 |
| HGF | 0.60 | 1.66E-05 | Unclustered | 0 | 4 | 0.011035 |
| HSPA1L | 0.99 | 8.11E-05 | Unclustered | 0 | 2 | 0.010692 |
| IFNAR2 | 0.52 | 3.25E-04 | Unclustered | 1 | 2 | 0 |
| IFNGR1 | −0.39 | 9.47E-04 | Unclustered | 0.5 | 4 | 0.005006 |
| IFNGR2 | 2.37 | 3.78E-08 | Unclustered | 1 | 2 | 0 |
| IKBKG | 0.64 | 4.51E-04 | Cluster 4 | 0.6 | 5 | 0.00319 |
| IL10RA | 0.80 | 5.84E-12 | Unclustered | 1 | 2 | 0 |
| IL16 | 0.38 | 0.001091 | Unclustered | 0 | 0 | 0 |
| IL18 | 0.56 | 8.19E-05 | Unclustered | 0.166667 | 4 | 0.033162 |
| IL21R | 1.04 | 3.15E-04 | Unclustered | 0.666667 | 3 | 0.001452 |
| IL2RB | 1.33 | 6.03E-04 | Unclustered | 1 | 2 | 0 |
| IL32 | −0.28 | 1.87E-05 | Unclustered | 0 | 1 | 0 |
| IL33 | 0.93 | 2.77E-09 | Unclustered | 1 | 2 | 0 |
| IL37 | 2.00 | 0.001238 | Unclustered | 0 | 1 | 0 |

*(Continued)*

**Table 2.** (Continued)

| SYMBOL | log2FC | p | Cluster | Clustering co-efficient | Degree | Betweeness Centrality |
|---|---|---|---|---|---|---|
| IL7 | 0.55 | 1.65E-06 | Unclustered | 0.3 | 5 | 0.004205 |
| IL7R | 0.56 | 1.93E-10 | Unclustered | 0.333333 | 3 | 0.01615 |
| INHBA | 0.72 | 4.68E-04 | Unclustered | 0 | 1 | 0 |
| IRAK1 | 0.64 | 1.03E-04 | Unclustered | 0.5 | 4 | 0.005121 |
| ITGA2 | 0.50 | 1.43E-06 | Cluster 2 | 0.464286 | 8 | 0.022655 |
| ITGA4 | 1.62 | 2.84E-08 | Cluster 2 | 0.666667 | 7 | 0.004541 |
| ITGAL | 0.52 | 1.69E-04 | Cluster 2 | 0.5 | 8 | 0.004652 |
| ITGAM | 2.33 | 1.17E-09 | Unclustered | 0.666667 | 4 | 0.003024 |
| ITGAX | 0.72 | 1.61E-04 | Unclustered | 1 | 3 | 0 |
| ITGB2 | 1.28 | 4.79E-14 | Cluster 2 | 0.418182 | 11 | 0.026134 |
| ITGB7 | −1.47 | 6.25E-05 | Cluster 2 | 0.619048 | 7 | 0.05336 |
| JAK1 | 0.37 | 4.40E-05 | Unclustered | 0.181818 | 12 | 0.071918 |
| KNG1 | 0.95 | 8.84E-04 | Unclustered | 0 | 0 | 0 |
| LAMA2 | 0.57 | 3.87E-06 | Cluster 8 | 1 | 2 | 0 |
| LAMB1 | 0.64 | 3.91E-04 | Cluster 8 | 1 | 2 | 0 |
| LAMB3 | 0.83 | 1.46E-04 | Unclustered | 0 | 1 | 0 |
| LAMC1 | 0.91 | 1.92E-05 | Cluster 8 | 1 | 2 | 0 |
| LAMC2 | 2.19 | 2.38E-09 | Unclustered | 0 | 1 | 0 |
| LPAR1 | −2.54 | 3.85E-08 | Cluster 9 | 1 | 2 | 0 |
| LTB | 0.84 | 1.19E-08 | Unclustered | 0 | 0 | 0 |
| MAGI2 | −0.48 | 1.20E-05 | Unclustered | 1 | 2 | 0 |
| MAP4K1 | 0.88 | 2.60E-06 | Unclustered | 0 | 1 | 0 |
| MAPK13 | −0.50 | 3.44E-06 | Cluster 6 | 0.2 | 5 | 0.047386 |
| MAPK8 | −0.38 | 7.42E-05 | Cluster 6 | 0.055556 | 9 | 0.158478 |
| MAPK8IP2 | 0.46 | 0.001162 | Unclustered | 0 | 2 | 0.015873 |
| MET | 0.76 | 1.28E-04 | Unclustered | 0.111111 | 10 | 0.17753 |
| MMP2 | 1.00 | 2.10E-04 | Unclustered | 0 | 4 | 0.033256 |
| MSN | 1.30 | 4.10E-08 | Unclustered | 0.333333 | 3 | 0.02271 |
| MYC | 0.90 | 4.42E-05 | Unclustered | 0.2 | 5 | 0.050895 |
| MYD88 | −0.41 | 8.80E-10 | Unclustered | 0.333333 | 7 | 0.041756 |
| MYL10 | 0.63 | 4.08E-04 | Unclustered | 0 | 1 | 0 |
| MYL12B | −0.63 | 6.94E-04 | Unclustered | 0 | 1 | 0 |
| NGF | −1.20 | 4.10E-07 | Unclustered | 0 | 0 | 0 |
| NLK | 0.33 | 1.06E-04 | Unclustered | 0 | 0 | 0 |
| PAK2 | −0.42 | 4.08E-04 | Unclustered | 0 | 2 | 0.003293 |
| PAK4 | 0.40 | 6.55E-04 | Unclustered | 1 | 2 | 0 |
| PAK6 | 0.64 | 3.65E-04 | Unclustered | 1 | 2 | 0 |
| PDGFRA | 0.69 | 3.06E-06 | Unclustered | 0.4 | 5 | 0.006608 |
| PF4V1 | 1.87 | 2.60E-05 | Unclustered | 0.8 | 5 | 3.60E-05 |
| PFN1 | −0.36 | 2.95E-09 | Unclustered | 1 | 2 | 0 |
| PHLPP1 | −0.44 | 4.67E-05 | Unclustered | 0 | 0 | 0 |
| PIK3CB | −0.57 | 6.07E-05 | Unclustered | 0.186813 | 14 | 0.109273 |
| PIP5K1C | 1.52 | 3.66E-06 | Unclustered | 0.2 | 6 | 0.050518 |
| PLA2G4A | 0.73 | 2.38E-09 | Unclustered | 0 | 1 | 0 |
| PLCB4 | −1.09 | 3.17E-04 | Unclustered | 0 | 2 | 0.001472 |

*(Continued)*

**Table 2.** (Continued)

| SYMBOL | log2FC | p | Cluster | Clustering co-efficient | Degree | Betweeness Centrality |
|---|---|---|---|---|---|---|
| PLCG2 | 1.51 | 4.65E-04 | Cluster 5 | 0.222222 | 9 | 0.068073 |
| PRKCB | 0.60 | 1.74E-07 | Unclustered | 0 | 5 | 0.062154 |
| PTEN | 1.09 | 4.16E-06 | Unclustered | 0.190476 | 7 | 0.030663 |
| RAC2 | 0.39 | 8.47E-08 | Cluster 5 | 0.111111 | 10 | 0.082098 |
| RASA2 | −0.77 | 2.32E-06 | Unclustered | 0 | 1 | 0 |
| RDX | 0.58 | 7.86E-04 | Unclustered | 1 | 2 | 0 |
| RPS6 | 0.89 | 4.08E-04 | Unclustered | 0 | 1 | 0 |
| RPS6KA1 | 0.60 | 1.41E-04 | Unclustered | 0 | 1 | 0 |
| RPS6KA3 | −0.61 | 1.95E-05 | Unclustered | 0 | 1 | 0 |
| RXRA | 0.43 | 9.86E-06 | Unclustered | 0 | 0 | 0 |
| SCIN | 0.68 | 7.48E-04 | Unclustered | 0 | 1 | 0 |
| SNW1 | 1.48 | 2.85E-06 | Unclustered | 0 | 0 | 0 |
| SPP1 | 0.66 | 9.67E-07 | Unclustered | 1 | 2 | 0 |
| SSH1 | 0.55 | 1.46E-04 | Unclustered | 0 | 0 | 0 |
| STAT3 | 0.77 | 5.98E-04 | Unclustered | 0.127273 | 11 | 0.167661 |
| SYK | −0.40 | 1.01E-08 | Cluster 5 | 0.5 | 5 | 0.004511 |
| TCL1B | 3.42 | 4.39E-04 | Unclustered | 0 | 0 | 0 |
| THBS2 | 0.93 | 2.12E-12 | Unclustered | 0 | 0 | 0 |
| TLR2 | 0.33 | 2.49E-09 | Unclustered | 0.133333 | 6 | 0.069196 |
| TMSB4Y | 1.68 | 2.59E-04 | Unclustered | 0 | 1 | 0 |
| TNC | 1.11 | 1.89E-13 | Unclustered | 0 | 0 | 0 |
| TNFRSF11B | 1.96 | 1.54E-04 | Unclustered | 0 | 1 | 0 |
| TNFRSF17 | 0.65 | 2.24E-05 | Unclustered | 0 | 0 | 0 |
| TNFRSF1A | 0.93 | 6.06E-05 | Cluster 4 | 0.333333 | 6 | 0.01659 |
| TNFRSF1B | 1.14 | 2.62E-07 | Unclustered | 1 | 2 | 0 |
| TNFSF10 | 0.91 | 1.21E-06 | Unclustered | 0.333333 | 3 | 0.015873 |
| TNFSF15 | 0.85 | 4.47E-08 | Unclustered | 0 | 0 | 0 |
| VAV1 | 1.37 | 5.47E-06 | Cluster 5 | 0.533333 | 6 | 0.007671 |
| VCAM1 | −1.66 | 6.39E-07 | Cluster 2 | 0.254545 | 11 | 0.125962 |
| VWF | 1.73 | 1.37E-08 | Unclustered | 0 | 2 | 0.00994 |
| XCL1 | 0.42 | 7.87E-05 | Unclustered | 0.866667 | 6 | 2.90E-04 |
| YWHAH | 1.02 | 3.82E-06 | Unclustered | 0 | 1 | 0 |

analysis on the dataset revealed that PI3K-AKT pathway is highly relevant in the pathogenesis of DN. The analysis also highlighted that inflammatory signaling pathways, and ECM related pathways were enriched in DN condition, with 156 unique genes contributing to these pathways. Additionally, 21 genes involved in AGE-RAGE signaling and 3 genes involved with Notch signaling were also identified. AGE-RAGE interaction is known to trigger oxidative stress in renal tissues by increasing ROS levels, which sets the wheel in motion for PI3K-AKT signaling [33]. Together, we proceeded to look into these 167 genes to understand the reactivation mechanism.

Prioritizing the 167 genes we created a protein interaction network via STRING app of Cytoscape. Among them 58 showed high connectivity based on their topological properties such as degree, betweenness centrality, and clustering coefficient (Table 2). Hence, we created a sub-network with this condensed interaction. This core was reanalyzed for highlighting the core pathways. Apart from the previous mentioned pathways the core was associated with Chemokine

**Table 3. 58 genes from the condensed network.**

| ACKR4 | CCR5 | CXCR4 | IFNGR1 | ITGAM | PF4V1 |
|---|---|---|---|---|---|
| ACTN1 | CCR6 | CXCR6 | IFNGR2 | ITGAX | PFN1 |
| ACTR3 | COL1A2 | DUSP1 | IKBKG | ITGB7 | RDX |
| ARPC1B | COL3A1 | ENAH | IL10RA | LAMA2 | SPP1 |
| ARPC3 | COL6A2 | FGF7 | IL21R | LAMB1 | SYK |
| CCL19 | COL6A3 | GADD45A | IL2RB | LAMC1 | TNFRSF1B |
| CCL2 | CX3CR1 | GADD45B | IL33 | LPAR1 | VAV1 |
| CCL21 | CXCL1 | GADD45G | IRAK1 | MAGI2 | XCL1 |
| CCNE1 | CXCL6 | GNA13 | ITGA4 | PAK4 | |
| CCR2 | CXCL8 | IFNAR2 | ITGAL | PAK6 | |

signaling pathway, and NF-kB signaling pathway. PI3K-AKT signaling [33] induced by AGE-RAGE interaction can activate NF-kB [55].

In our dataset CCNE1, COL1A2, COL6A2, COL6A3, FGF7, IFNAR2, IKBKG, IL2RB, ITGA4, ITGB7, LAMA2, LAMB1, LAMC1, LPAR1, MAGI2, SPP1, and SYK were associated with PI3K-AKT signaling. COL6A2 and ITGB7 showed similar expression pattern in both independent and existing dataset. Most of these differentially expressed genes were responsible for focal adhesion and regulation of actin cytoskeleton, thereby justifying enrichment of the EMT hallmark from GSEA, except for IFNAR2, IKBKG, and IL2RB, which were associated with inflammation.

Down regulation of laminins (LAMA2, LAMB1, and LAMC1) and integrins (ITGA4, ITGAL, ITGAX, and ITGB7) were observed in our dataset. LAMB1 (log2FC = .95, p = 0.00039) was negatively correlated to IKBKG ($\beta = -0.8$) and, IL21R ($\beta = -0.05$), ITGAX ($\beta = -0.06$), MAGI2 ($\beta = -0.33$) and PAK4 ($\beta = -0.60$). LAMA2 (log2FC = .37, p = 3.87E-06), a key molecule in the PI3K-AKT pathway, is known to activate cell migration [56]. Down regulation of laminins and integrins are associated with modulation of focal adhesion, which is also corroborated from KEGG analysis [57]. CCNE1 and MAGI2 were unique to PI3K-AKT signaling. All these genes, except COL1A2, COL6A3, and SPP1 were downregulated. Interaction of RAGE with AGEs is linked with PI3K-AKT signaling. Reports suggest RAGE is a multiligand receptor that activates PI3K-AKT signaling but upon interaction with AGEs leads to inhibition of the signaling, which induces autophagy. Administration of RAGE neutralizing antibody, PI3K/AKT signaling agonist and/ or AKT inhibitors can revert the inhibition [58–60]. Down regulation of CCNE1 (log2FC = 0.48, p = 0.000140249) indicated dysregulation in podocyte phenotype [61]. MAGI2 (log2FC = −2.54, p = 1.20E-05) has been reported to have a protective role in the podocytes and its down regulation indicates a decrease in anti-apoptotic proteins and is correlated to podocyte loss [62]. Just like Notch, COL1A2 (log2FC = 3.34, p = 1.12E-14) is active during developmental stages but gets attenuated in adults [63]. COL6A3 (log2FC = 3.38, p = 7.94E-13) and SPP1(log2FC = 1.48, p = 9.67E-07) have been proposed to be profibrotic genes that participate in immunologic modulation. The upregulation of these genes indicates kidney injury [64,65]. Down regulation of the rest of the genes due to low PI3K-AKT signaling led to dysregulation of focal adhesion and regulation of cytoskeleton. Chronic hypergylcemia result in down regulation of PI3K-AKT signaling which is evident from the dysregulation in the expression profiles of the genes involved with the signaling. Since PI3K-AKT signaling is a survival pathway, its down regulation in DN condition manifests as podocytopathy. Findings from biopsies of FSGS patients have also shown decreased expression of pSer473-Akt, showing progression to end stage kidney disease [66].

The network evidently showed inflammatory signals were the major drivers in DN pathogenesis. The inflammatory markers, some of which are also involved with NF-kB signaling, are positively correlated amongst themselves as well as with certain ECM and cytoskeletal markers (Fig 4C) indicating their tight regulation. A previous report from our lab also showed exposure of renal cells with N-carboxymethyl-lysine (CML), a predominant type of AGEs, induced Zeb2, a transcription factor of EMT via NF-kB signaling, thereby affecting podocyte integrity [67]. Genes viz IKBKG (log2FC = −0.38,

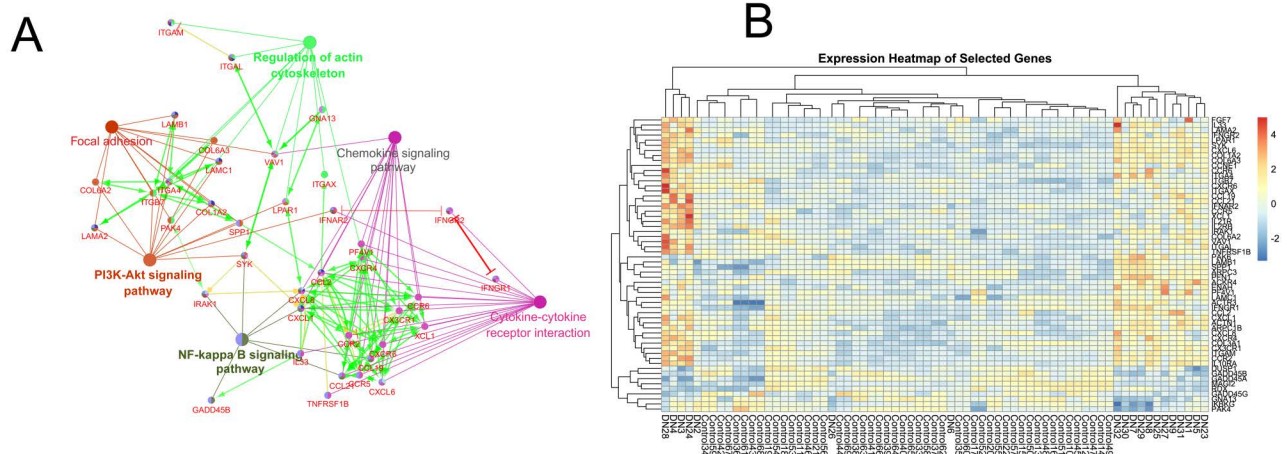

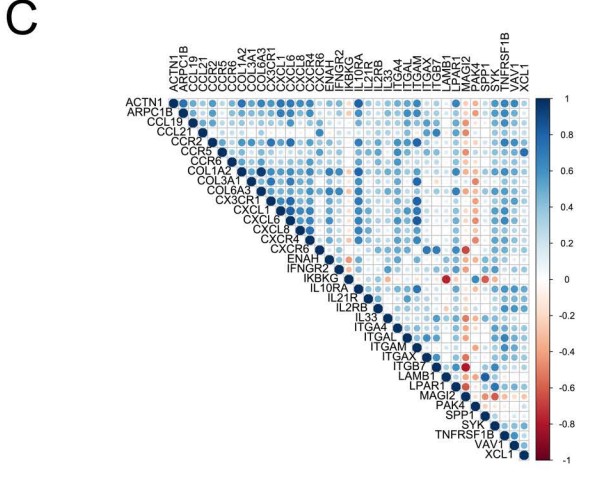

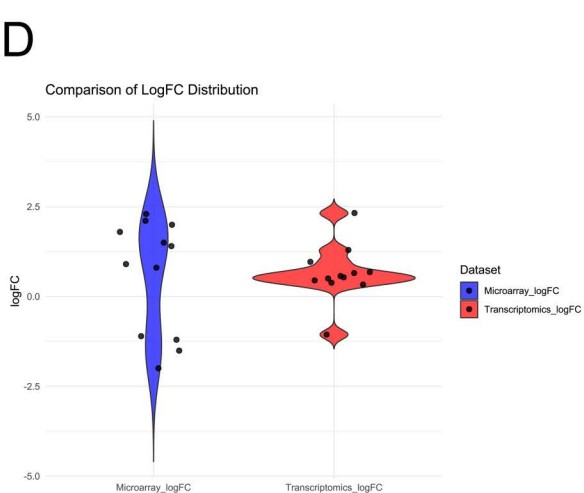

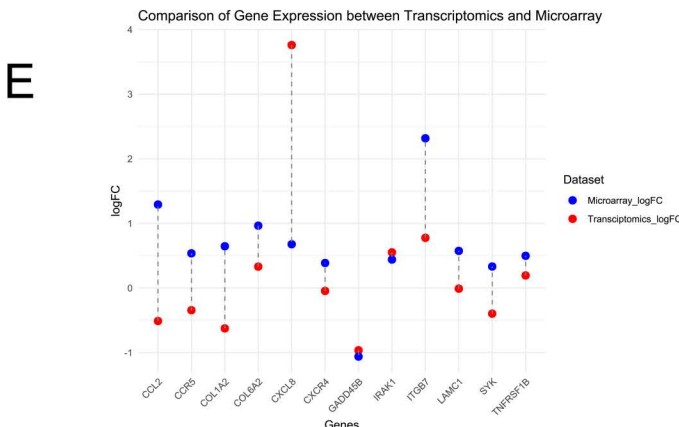

**Fig 4. Chemokine markers, ECM/ fibrotic markers and stress response markers were the central players.** A. Network shows the interaction between the 58 core genes and the pathways. Edges in green, yellow and red depict activation, expression and inhibition respectively. The network depicts 3 major clusters- inflammatory cluster (CXCR4, CCR5, CXCL6, CCL19, TNFRSF1B, IKBKG, IRAK1), signaling cluster (PI3K-AKT signaling), and fibrosis cluster (COL1A2, COL6A3, LAMB1, ITGB7). B. Heatmap shows differential expression of the 58 genes in the condensed network based on their expression. Most of the genes were positively correlated except for IKBKG, LAMB1, MAGI2, and PAK4 suggesting that inflammation and ECM genes are tightly regulated together. C. Correlation plot highlight 27 genes whose p values were ≤ 1e-14. The upregulated genes (depicted in red) point at their potential involvement in the disease progression and may serve as biomarkers or therapeutic targets for AGE induced DN, while downregulated genes (in blue) indicate alteration and suppression in the signaling of the regulatory mechanism. D. The violin plot shows the distribution of expressional fold changes of the 12 driver genes across the microarray dataset (existing dataset) (blue violin) and regional transcriptomic dataset (independent dataset) (red violin) from KPMP public database. E. The scatter plot with connected lines shows difference in expression in the two datasets. The blue dots represent expression values of existing dataset and the red dot depicts expression values of independent dataset. The graph shows consistent pattern of expression in COL6A2, ITGB7, IRAK1, TNFRSF1B, GADD45B. CXCL8 is shows upregulation in both sets but the long connecting line indicates difference in the expression level.

p = .00045) and SYK (log2FC=0.77, p = 1.01E-08) both of which participated in both NF-kB signaling and PI3K-AKT signaling were found to downregulated. IKBKG was negatively correlated with most of the other genes ($\beta_{COL1A2}$ = −0.27, $\beta_{COL6A3}$ = −0.33, $\beta_{CX3CR1}$ = −0.21). Both these genes are crucial for mediating NF-kB signaling, which is a key cell survival pathway and their down regulation leads to fibrosis. In steatohepatitis knockdown of IKBKG resulted in fibrosis, while inhibition of SYK decreased fibrosis in kidney [68,69]. Upregulation was observed in chemokine and cytokine regulating genes (ACKR4 (log2FC = 1.18, p = 6.12E-11), CCL19 (log2FC 2.84, p = 6.86E-10), CCL2 (log2FC = 1.86. p = 9.33E-09), CCR2 (log2FC = 1.45, p = 6.43E-12), CX3CR1 (log2FC = 1.97, p = 2.19E-07), CXCL1 (log2FC = 1.60, p = 6.72E-06), CXCL6 (log2FC = 4.66, p = 3.44E-17), CXCR4 (log2FC = 1.39, p = 1.92E-07), IL10RA (log2FC = 2.37, p = 5.84E-12), and IL33 (log2FC = 1.33, p = 2.77E-09)) indicating an increase in recruitment and activation of inflammatory pathways [70]. The differential levels of cytokines and chemokines boost inflammation which affects ECM remodeling as evident from the expression profiles of the gene participating in focal adhesion and regulation of actin cytoskeleton. These levels of cytokines and chemokines lead to constitutive activation of NF-kB signaling which can activate Notch signaling [71]. This is known to exacerbate podocyte injury, a classic hallmark of DN insult [72]. Damaged kidney cells recruit macrophages that are polarized into the proinflammatory M1 phenotype and participate in the death of intrinsic kidney cells. The crosstalk between Notch and key NF-kB molecules favors this polarization and aggravates the tissue damage, providing a new direction of therapeutic target for diabetic nephropathy in the future [73].

Our cross-platform results identified a condensed AGE-associated network with several inflammation and ECM-related mediators. Importantly, DDIT3, GADD45A, THBS2, CCL2 and CSF1R were consistently altered across microarray and RNA-seq datasets, implicating stress (DDIT3), DNA damage/repair stress pathways (GADD45A), matricellular remodeling (THBS2), monocyte recruitment (CCL2) and macrophage signaling (CSF1R) in AGE-induced perturbations of renal cells. Although DDIT3, THBS2, and CSF1R were not part of the sub-network of 58 genes and not discussed in detail above. Recent studies corroborate roles for DDIT3 in renal fibrosis and ER stress-driven injury [74]. Interestingly, in both our datasets DDIT3 is downregulated. Reports by Kong et al. suggest upregulation of DDIT3 promotes fibrosis in diabetic kidney whereas report by Li-Li Ma et al. suggest DDIT3 is downregulated in DN condition [75]. This may be an adaptive response to attenuate persistent ER stress rather. GADD45A/B, THBS2 are identified as biomarkers and functional modulators in acute renal injury and DKD [76–78]. The centrality of CCL2 has been discussed above. These reports place our findings within the contemporary literature and highlight the translational potential of targeting inflammation/macrophage and matricellular remodeling pathways in AGE-driven DN.

Collectively our study showed that AGEs fostered by chronic hyperglycemia suppress PI3K-AKT pathway, activate NF-kB signaling and elicit inflammatory responses. Both these responses have been shown to reactivate Notch signaling [79,80]. On the other hand, reactivation of Notch also suppresses PI3K-AKT signaling. This induces apoptosis in the terminally differentiated podocytes [81,82] (Fig 6).

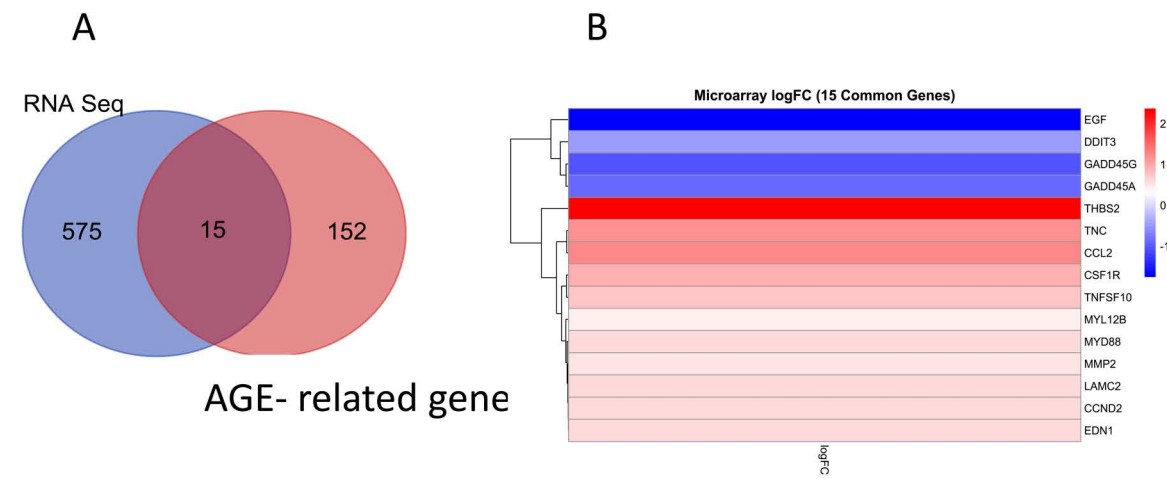

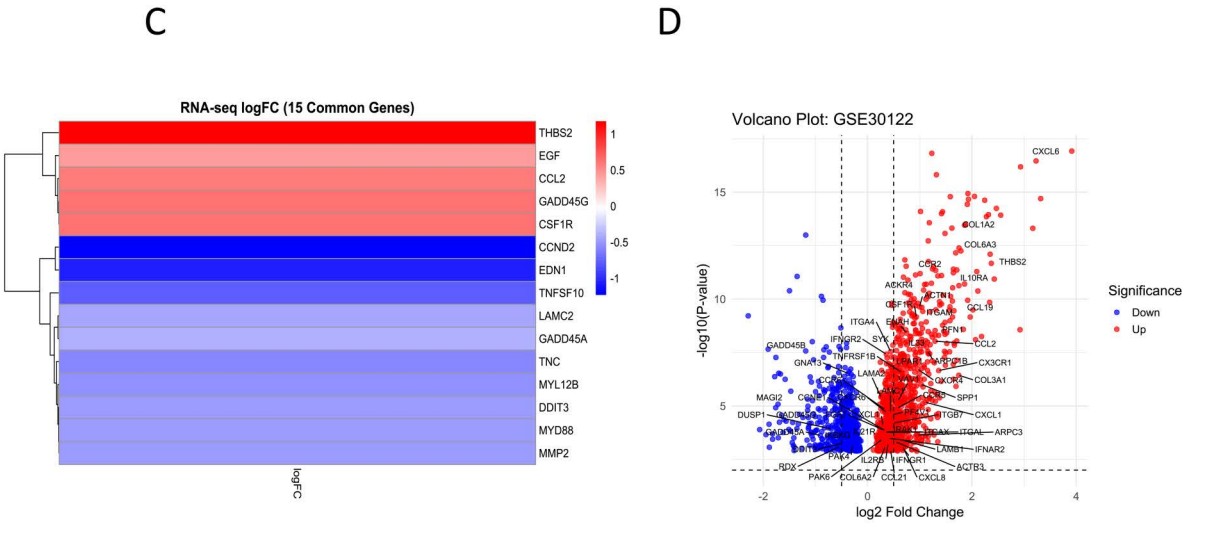

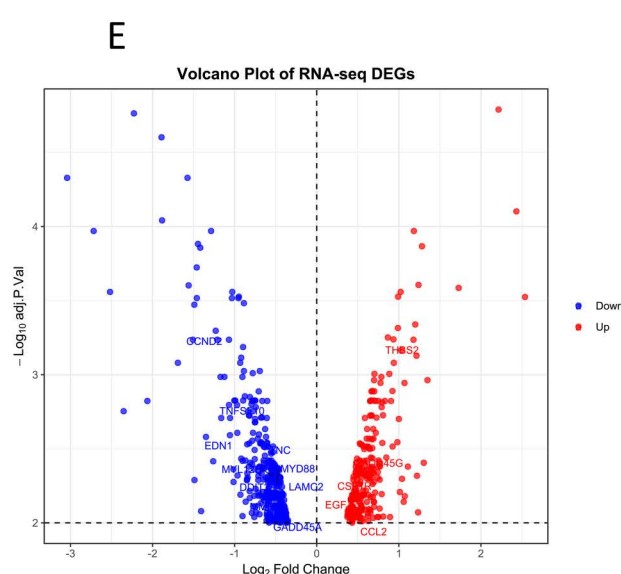

**Fig 5. 15 genes were found to be common upon cross-platform validation with RNA-Seq data.** A. Venn diagram representing 15 genes are common between signifiant RNA seq dataset and the 167 genes related to AGEs obtained from microarray dataset. B: Heatmap representing the logFC of expression values 15 genes from microarray dataset. C. Heatmap representing the logFC of the quantified transcript expression values 15 genes from RNA Seq dataset. DDIT3, GADD45A, THBS2, CCL2, and CSF1R showed consistent expression trends across platforms. D. Volcano plot showing the differentially expressed genes from microarray dataset GSE30122. 58 core genes from the condensed gene network along with the 5 genes that were consistently altered across microarray and RNA-seq dataset are highlighted. Genes significantly upregulated are shown in red, while significantly downregulated genes are shown in blue. E. Volcano plot showing the differentially expressed genes from RNA-seq dataset GSE299230. 15 genes from the condensed gene network from microarray dataset GSE30122 and RNA-seq dataset GSE299230 including the 5 genes that were consistently altered across microarray and RNA-seq dataset are highlighted. Genes significantly upregulated are shown in red, while significantly downregulated genes are shown in blue.

**Table 4. List of the common genes between microarray dataset and RNA Seq dataset.**

| Common genes |
| --- |
| MMP2 |
| EDN1 |
| TNC |
| MYD88 |
| GADD45G |
| MYL12B |
| GADD45A |
| DDIT3 |
| TNFSF10 |
| THBS2 |
| EGF |
| LAMC2 |
| CCND2 |
| CSF1R |
| CCL2 |

## Conclusion

The study provides novel information about AGEs induced podocyte injury. AGEs induced selective network of 58 genes in podocytes and are associated with non-canonical reactivation of Notch signaling, probably via activating NF-kB signaling and supressing PI3K-AKT signaling. These signaling cascades that are selectively induced in podocytes under the stimulus of AGEs could be explored as therapeutic targets to prevent the progression of diabetic kidney disease. Our earlier experimental studies also suggest reactivation of Notch signaling is implicated in the podocytes [11].

## Limitations

The public RNA-seq dataset used for validation (GSE299230) is an in vitro HK-2 cell model (hnRNPF overexpression; n = 3 per group) rather than human patient tissue; the small sample size and cellular model limit direct generalizability to complex human kidney disease. Second, for comparisons across platforms (microarray GSE30122 vs RNA-seq GSE299230), platform-specific biases (probe design, dynamic range, normalization) can influence observed concordance; we therefore used conservative thresholds and variance-shrinkage methods and present GSE299230 results as supportive, not definitive, evidence. Finally, our study is computational and correlative; experimental validation (knockdown/overexpression, protein-level assays, in vivo models) is necessary to confirm the mechanistic roles of condensed gene network in AGE-driven podocyte/renal injury.

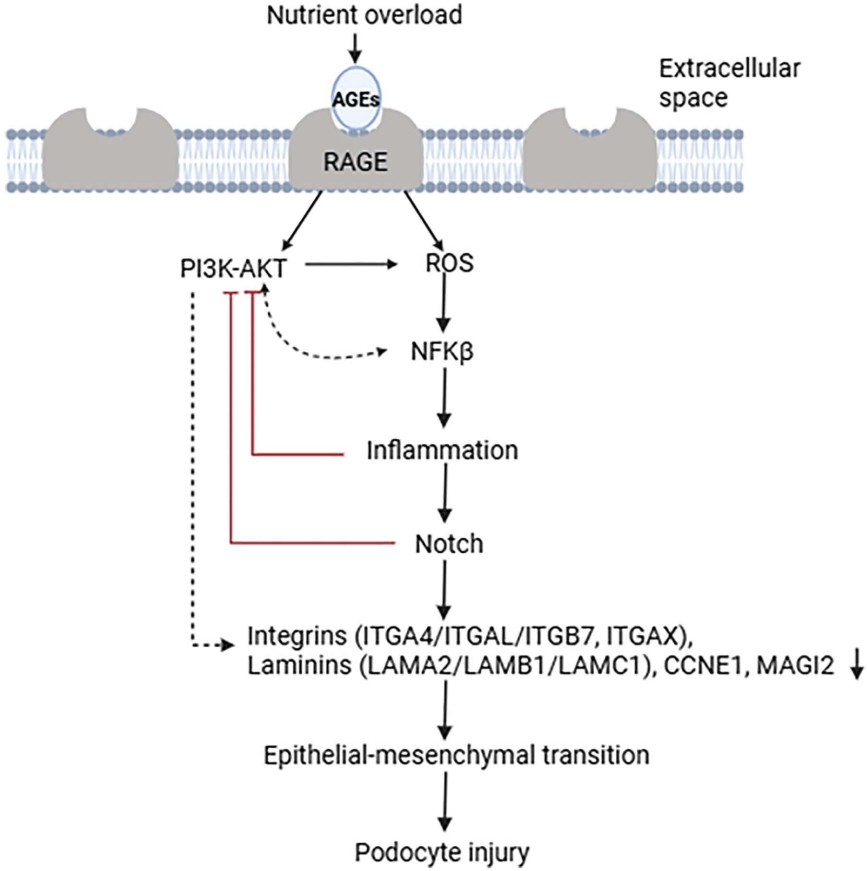

**Fig 6. Interaction between AGE-RAGE induces ROS production and typically sets on PI3K-AKT pathway.** The simultaneous influence of both activates the NFKB signaling which elicits inflammatory responses. The induction of the NFKB signaling also triggers NICD1, thus leading to non-cannonical activation of the Notch signaling pathway. This reactivation suppresses PI3K-AKT signaling and disregulates the expression of integrins (ITGA4, ITGAL, ITGB7, ITGAX) and laminins (LAMA2, LAMB1, LAMC1), along with structural proteins like CCNE1 and MAGI2, leading to ECM remodeling and changes in cell adhesion and migration. These molecular changes contribute to EMT, where podocytes lose their epithelial characteristics and acquire mesenchymal-like properties, ultimately leading to podocyte injury.

## Supporting information

**S1 File. S1 Table.** List of up and down regulated genes from GSE 30122. **S2 Table.** KEGG Pathways associated with DEGs. **S3 Table.** Hallmarks in the dataset obtained from GSE analysis. **S4 Table.** Trend for logFC within the existing data and independent data from KPMP Database.
(ZIP)

## Author contributions

**Conceptualization:** Somorita Baishya, Pramod R. Somvanshi.

**Data curation:** Somorita Baishya, Adyasha Sarangi.

**Formal analysis:** Somorita Baishya.

**Investigation:** Somorita Baishya, Pramod R. Somvanshi.

**Methodology:** Somorita Baishya.

**Supervision:** Pramod R. Somvanshi, Anil Kumar Pasupulati.

**Validation:** Somorita Baishya.

**Visualization:** Somorita Baishya.

**Writing – original draft:** Somorita Baishya.

**Writing – review & editing:** Pramod R. Somvanshi, Anil Kumar Pasupulati.

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
