## [Decision Letter · Decision Letter 0]

18 Jul 2025

Dear Dr. Pasupalati,

Thank you for submitting your manuscript to PLOS ONE. After careful consideration, we feel that it has merit but does not fully meet PLOS ONE’s publication criteria as it currently stands. Therefore, we invite you to submit a revised version of the manuscript that addresses the points raised during the review process.

We look forward to receiving your revised manuscript.

Kind regards,

Vinay Randhawa, Ph.D.

Academic Editor

PLOS ONE

**Journal Requirements:**

1. When submitting your revision, we need you to address these additional requirements. Please ensure that your manuscript meets PLOS ONE's style requirements, including those for file naming. The PLOS ONE style templates can be found at https://journals.plos.org/plosone/s/file?id=wjVg/PLOSOne_formatting_sample_main_body.pdf and https://journals.plos.org/plosone/s/file?id=ba62/PLOSOne_formatting_sample_title_authors_affiliations.pdf 2. Thank you for stating the following in the Acknowledgments Section of your manuscript: The authors acknowledge the support of the DBT-RA Program (DBT-RA/2023/ January/NE/3723). We note that you have provided funding information that is not currently declared in your Funding Statement. However, funding information should not appear in the Acknowledgments section or other areas of your manuscript. We will only publish funding information present in the Funding Statement section of the online submission form. Please remove any funding-related text from the manuscript and let us know how you would like to update your Funding Statement. Currently, your Funding Statement reads as follows: The author(s) received no specific funding for this work.  Please include your amended statements within your cover letter; we will change the online submission form on your behalf. 3. PLOS requires an ORCID iD for the corresponding author in Editorial Manager on papers submitted after December 6th, 2016. Please ensure that you have an ORCID iD and that it is validated in Editorial Manager. To do this, go to ‘Update my Information’ (in the upper left-hand corner of the main menu), and click on the Fetch/Validate link next to the ORCID field. This will take you to the ORCID site and allow you to create a new iD or authenticate a pre-existing iD in Editorial Manager. 4. Please include captions for your Supporting Information files at the end of your manuscript, and update any in-text citations to match accordingly. Please see our Supporting Information guidelines for more information: http://journals.plos.org/plosone/s/supporting-information. 5. If the reviewer comments include a recommendation to cite specific previously published works, please review and evaluate these publications to determine whether they are relevant and should be cited. There is no requirement to cite these works unless the editor has indicated otherwise. 

**Additional Editor Comments:**

Thank you for submitting your manuscript to PLOS ONE. We have now received detailed feedback from the reviewers and invite you to submit a revised version of your manuscript. Please address each of the points outlined below in a point‐by‐point response, indicating how and where in the revised text each concern has been addressed.

3. Editor’s Comments

Major Comments

1. Both reviewers have noted the abundant availability of next-generation sequencing (NGS) datasets with greater dynamic range and sensitivity than microarrays. Please provide a concise justification for electing to analyze a microarray dataset rather than RNA-Seq or other high-throughput platforms. Importantly, the manuscript would be strengthened by an independent validation of your key findings using RNA-Seq data.

2. Clearly articulate the novel aspects of this study in the context of existing cardiovascular transcriptomics research. How do your findings advance the field beyond currently published microarray and RNA-Seq analyses?

3. Update the Introduction and Discussion to include and discuss the most recent (past 2–3 years) publications relevant to transcriptomic profiling in atherosclerosis. This will ensure that your work is situated appropriately within the current state of the art

Minor Comments

1. Please annotate the heatmap in Figure 2 with the names of the top up- and down-regulated genes for clarity.

2. To better visualize how the 167 differentially expressed genes map onto the top enriched pathways, consider adding a heatmap or chord diagram illustrating their distribution.

3. Figure 1 requires improved layout and resolution. Please enhance the graphical design (e.g., consistent fonts, clearer panel labels, higher-resolution images) to meet journal quality standards.

4. The current heatmap in Figure 4B is limited in scope. I recommend displaying all samples (rather than a subset) to provide a more comprehensive view of gene expression patterns across experimental groups.

Reviewer 1

The manuscript presents a compelling in silico analysis exploring gene regulatory networks associated with Notch signaling activation by AGEs in the pathogenesis of diabetic kidney disease (DKD). While the study is well-conceived, I have a few suggestions and points for clarification before the manuscript proceeds to publication:

• Authors have performed microarray data analysis for studying diabetic nephropathy (DN) and diabetic kidney disease (DKD). Given that RNA-Seq is a more advanced and comprehensive technique for transcriptomic profiling, could you please elaborate on the rationale for choosing microarrays over RNA-Seq? Was this decision influenced by data availability, platform compatibility, or any other specific reason?

• Authors have employed a combination of bioinformatics tools for functional enrichment analysis, Gene set enrichment analysis, and PPI network, which is commendable. However, I noticed that references for these tools and methods have not been provided. Could you please include appropriate citations for the tools and databases used, to ensure clarity and reproducibility for readers?

• I noticed that some of the figure captions are quite lengthy and include detailed methodological descriptions. I would recommend revising them to be more concise and focused on explaining the figure content clearly. Methodological details can be moved to the main Methods section to improve readability and maintain consistency.

• In the conclusion, the authors mention (“Our earlier experimental studies also suggest reactivation of Notch signaling is implicated in the podocytes”) that their earlier experimental studies suggest reactivation of Notch signaling in podocytes. However, I could not find any experimental validation results or supporting data presented in the current manuscript. If such validation has been performed previously, it would be helpful to include the relevant citation or briefly summarize the findings for context and clarity.

Reviewer 2

Although the study is well-structured and based on the GSE30122 microarray dataset, the authors do not provide an explicit justification for the exclusive use of this platform, particularly considering the availability of public RNA-Seq datasets directly related to diabetic kidney disease.

I strongly recommend that the authors consult the NCBI Sequence Read Archive (SRA), which contains relevant transcriptomic data from human and murine podocytes under AGE-induced stress (e.g., SRX29078977 and SRX27070947). Given that this is an in silico study, the analysis is expected to be technically comprehensive, and a comparison with RNA-Seq data should be considered a fundamental step, either to cross-validate differentially expressed genes or to reinforce the robustness of the proposed regulatory network.

If such a comparison is not made, I suggest that its absence be explicitly addressed and discussed as a limitation of the study.

Reviewers' comments:

Reviewer's Responses to Questions

**Comments to the Author**

1. Is the manuscript technically sound, and do the data support the conclusions?

Reviewer #1: Yes

Reviewer #2: Partly

2. Has the statistical analysis been performed appropriately and rigorously?

Reviewer #1: Yes

Reviewer #2: I Don't Know

3. Have the authors made all data underlying the findings in their manuscript fully available?

Reviewer #1: Yes

Reviewer #2: No

4. Is the manuscript presented in an intelligible fashion and written in standard English?

Reviewer #1: Yes

Reviewer #2: Yes

**Reviewer #1:**  The manuscript presents a compelling in silico analysis exploring gene regulatory networks associated with Notch signaling activation by AGEs in the pathogenesis of diabetic kidney disease (DKD). While the study is well-conceived, I have a few suggestions and points for clarification before the manuscript proceeds to publication:

• Authors have performed microarray data analysis for studying diabetic nephropathy (DN) and diabetic kidney disease (DKD). Given that RNA-Seq is a more advanced and comprehensive technique for transcriptomic profiling, could you please elaborate on the rationale for choosing microarrays over RNA-Seq? Was this decision influenced by data availability, platform compatibility, or any other specific reason?

• Authors have employed a combination of bioinformatics tools for functional enrichment analysis, Gene set enrichment analysis, and PPI network, which is commendable. However, I noticed that references for these tools and methods have not been provided. Could you please include appropriate citations for the tools and databases used, to ensure clarity and reproducibility for readers?

• I noticed that some of the figure captions are quite lengthy and include detailed methodological descriptions. I would recommend revising them to be more concise and focused on explaining the figure content clearly. Methodological details can be moved to the main Methods section to improve readability and maintain consistency.

• In the conclusion, the authors mention (“Our earlier experimental studies also suggest reactivation of Notch signaling is implicated in the podocytes”) that their earlier experimental studies suggest reactivation of Notch signaling in podocytes. However, I could not find any experimental validation results or supporting data presented in the current manuscript. If such validation has been performed previously, it would be helpful to include the relevant citation or briefly summarize the findings for context and clarity.

**Reviewer #2:**  Dear Authors,

Although the study is well-structured and based on the GSE30122 microarray dataset, the authors do not provide an explicit justification for the exclusive use of this platform, particularly considering the availability of public RNA-Seq datasets directly related to diabetic kidney disease.

I strongly recommend that the authors consult the NCBI Sequence Read Archive (SRA), which contains relevant transcriptomic data from human and murine podocytes under AGE-induced stress (e.g., SRX29078977 and SRX27070947). Given that this is an in silico study, the analysis is expected to be technically comprehensive, and a comparison with RNA-Seq data should be considered a fundamental step, either to cross-validate differentially expressed genes or to reinforce the robustness of the proposed regulatory network.

If such a comparison is not made, I suggest that its absence be explicitly addressed and discussed as a limitation of the study.

Best regards,

**Do you want your identity to be public for this peer review?** For information about this choice, including consent withdrawal, please see our Privacy Policy

Reviewer #1: No

Reviewer #2: No

---

## [Author Response · Author response to Decision Letter 1]

3 Sep 2025

We appreciate the Editor and Reviewers for their time and insightful comments. We provided point-to-point responses to the reviewer’s comments, and we have revised the manuscript considering the comments from the reviewers. We acknowledge that the comments significantly improved the manuscript.

Editor’s Comments

Major Comments

1. Both reviewers have noted the abundant availability of next-generation sequencing (NGS) datasets with greater dynamic range and sensitivity than microarrays. Please provide a concise justification for electing to analyze a microarray dataset rather than RNA-Seq or other high-throughput platforms. Importantly, the manuscript would be strengthened by an independent validation of your key findings using RNA-Seq data.

Ans: The primary objective of the study was to investigate how AGEs evoke Notch signaling in glomerular podocytes and contribute to diabetic nephropathy (DN). For this purpose, we identified a microarray dataset (GSE30122) that is relevant to the pathophysiology of DN and provides an ample number of podocyte-specific samples for robust analysis. While we acknowledge that NGS platforms generally offer greater sensitivity and dynamic range, the availability of a robustly provided podocyte-specific RNA-seq dataset directly modeling AGE-mediated DN remains limited. During our search, other datasets were identified, however, those either involved different cell types, small sample sizes, or experimental conditions that did not satisfy our objective.

As suggested by the reviewers, we have analyzed the RNA Seq data of GSE299230 and found that similar trend in expression of DDIT3, GADD45A, THBS2, CCL2, and CSF1R between microarray and RNA-seq analysis. Notably, GSE299230 the experimental model involved human renal proximal tubular epithelial (HK-2) exposed to a hyperosmotic environment using 30 mM mannitol for 72 hours. Since, AGEs are formed through a slow non-enzymatic Maillard reaction between reducing sugars and free amino groups present in biological macromolecules. The conditions provided in GSE299230 do not appropriately reflect AGE mediated responses with respect to podocytes. Nevertheless, to address the reviewers concern we have included the comparative findings from GSE299230 in the supplementary file.

To further strengthen the robustness of our conclusions, we cross-validated our results with independent regional transcriptomic datasets available in the KPMP (Kidney Precision Medicine Project) database. Violin plot analysis (Figure 4D) showed the five genes (COL6A2, ITGB7, IRAK1, TNFRSF1B, and GADD45B) demonstrated consistent expression trends in both datasets, reinforcing their relevance in the context of AGE-induced DN (Figure 4E, Supplementary Table 4).

2. Clearly articulate the novel aspects of this study in the context of existing cardiovascular transcriptomics research. How do your findings advance the field beyond currently published microarray and RNA-Seq analyses?

Ans: The crux of the study is to unravel the mechanistic insights of influence of AGEs in podocytes, in the context of diabetic nephropathy. However, the effect of AGEs on cardiovascular complications is beyond the scope of this study.

3. Update the Introduction and Discussion to include and discuss the most recent (past 2–3 years) publications relevant to transcriptomic profiling in atherosclerosis. This will ensure that your work is situated appropriately within the current state of the art.

Ans: We have provided recent literature on how AGEs contribute to the micro and macrovascular complications in diabetic patients in the introduction section (references 21, 22). Since, the study focused on the influence of AGEs in eliciting reactivation of Notch signaling in the terminally differentiated podocytes, we did not cite the references related to transcriptomic profiling in atherosclerosis.

Minor Comments:

1. Please annotate the heatmap in Figure 2 with the names of the top up- and down-regulated genes for clarity.

Ans: We have provided the list of the top up-and down-regulated genes in supplementary table 4.

2. To better visualize how the 167 differentially expressed genes map onto the top enriched pathways, consider adding a heatmap or chord diagram illustrating their distribution.

Ans: We added a heatmap representing top 15 pathways and number of genes participating in each pathway as figure 3A and a detailed complete list has been provided in supplementary table 5.

3. Figure 1 requires improved layout and resolution. Please enhance the graphical design (e.g., consistent fonts, clearer panel labels, higher-resolution images) to meet journal quality standards.

Ans: We have enhanced the layout resolution of Figure1.

4. The current heatmap in Figure 4B is limited in scope. I recommend displaying all samples (rather than a subset) to provide a more comprehensive view of gene expression patterns across experimental groups.

Ans: We have incorporated the suggested changes in the Figure 4B of the revised manuscript.

Reviewer 1

1. Authors have performed microarray data analysis for studying diabetic nephropathy and diabetic kidney disease. Given that RNA-Seq is a more advanced and comprehensive technique for transcriptomic profiling, could you please elaborate on the rationale for choosing microarrays over RNA-Seq? Was this decision influenced by data availability, platform compatibility, or any other specific reason?

Ans: The primary objective of the study was to investigate how AGEs evoke Notch signaling in glomerular podocytes and contribute to diabetic nephropathy (DN). During our search for suitable transcriptomic datasets, we came across GSE30122, a microarray dataset, which is highly relevant to the pathophysiology of DN and provides an ample number of samples for robust analysis. Although other datasets were also identified, their limited sample sizes made them less suitable for our study.

2. Authors have employed a combination of bioinformatics tools for functional enrichment analysis, Gene set enrichment analysis, and PPI network, which is commendable. However, I noticed that references for these tools and methods have not been provided. Could you please include appropriate citations for the tools and databases used, to ensure clarity and reproducibility for readers?

Ans: In the revised manuscript, we have provided the appropriate references for the tools and databases used in the study.

3. I noticed that some of the figure captions are quite lengthy and include detailed methodological descriptions. I would recommend revising them to be more concise and focused on explaining the figure content clearly. Methodological details can be moved to the main Methods section to improve readability and maintain consistency.

Ans: In the revised manuscript we revised the figure captions for brevity and succinct.

4. In the conclusion, the authors mention (“Our earlier experimental studies also suggest reactivation of Notch signaling is implicated in the podocytes”) that their earlier experimental studies suggest reactivation of Notch signaling in podocytes. However, I could not find any experimental validation results or supporting data presented in the current manuscript. If such validation has been performed previously, it would be helpful to include the relevant citation or briefly summarize the findings for context and clarity.

Ans: Dear reviewer, our group has been pursuing studies with AGEs for the last ten years or so. We made several observations on cellular and molecular effects of AGEs on kidney function by performing wet-lab experiments (reference 9 (PMID: 32601154), reference 28 (PMID: 34277660), reference 6 (PMID: 25309512)). We recently shifted the focus to analyzing the transcriptomic data employing a few bioinformatics tools. Very interestingly, the outcome of bioinformatics analysis is redundant with our wet-lab studies, which we cited appropriately in the manuscript, and we highlighted in the revised manuscript.

Reviewer 2

1. Although the study is well-structured and based on the GSE30122 microarray dataset, the authors do not provide an explicit justification for the exclusive use of this platform, particularly considering the availability of public RNA-Seq datasets directly related to diabetic kidney disease.

I recommend that the authors consult the NCBI Sequence Read Archive (SRA), which contains relevant transcriptomic data from human and murine podocytes under AGE-induced stress (e.g., SRX29078977 and SRX27070947). Given that this is an in silico study, the analysis is expected to be technically comprehensive, and a comparison with RNA-Seq data should be considered a fundamental step, either to cross-validate differentially expressed genes or to reinforce the robustness of the proposed regulatory network.

If such a comparison is not made, I suggest that its absence be explicitly addressed and discussed as a limitation of the study.

Ans: We thank the reviewer for an excellent suggestion. The study's primary objective was to investigate how AGEs evoke Notch signaling in glomerular podocytes and contribute to diabetic nephropathy. During our search for suitable transcriptomic datasets, we came across GSE30122. This microarray dataset was highly relevant to the pathophysiology of diabetic nephropathy and provides an ample number of samples for robust analysis. Although other datasets have been identified, their limited sample sizes have made them less suitable for our study.

As suggested by the reviewers, we analyzed the RNA-seq data of GSE299230 and found a similar trend in expression of DDIT3, GADD45A, THBS2, CCL2, and CSF1R between microarray and RNA-seq analysis. Notably, GSE299230, the experimental model involved human renal proximal tubular epithelial (HK-2) cells exposed to a hyperosmotic environment using 30 mM mannitol for 72 hours. AGEs are formed through a slow non-enzymatic Maillard reaction between reducing sugars and free amino groups in biological macromolecules. The conditions provided in GSE299230 do not appropriately reflect AGE-mediated responses concerning podocytes. Nevertheless, to address the reviewers' concern, we have included the findings from GSE299230 in the supplementary file.

---

## [Editor Report · Decision Letter 1]

24 Sep 2025

Dear Dr. Pasupulati,

Thank you for submitting your manuscript to PLOS ONE. After careful consideration, we feel that it has merit but does not fully meet PLOS ONE’s publication criteria as it currently stands. Therefore, we invite you to submit a revised version of the manuscript that addresses the points raised during the review process.

We look forward to receiving your revised manuscript.

Kind regards,

Vinay Randhawa, Ph.D.

Academic Editor

PLOS ONE

Journal Requirements:

**Additional Editor Comments:**

The manuscript is promising but not yet ready for publication. Substantial additional computational analyses—especially inclusion of GSE299230 with focused evaluation of DDIT3, GADD45A, THBS2, CCL2, and CSF1R—plus updated literature and clearer figure/reporting standards are required to support the claims.

Main Concerns

1. The current analyses do not convincingly establish the role of the proposed core genes. Please add rigorous computational experiments.

2. Please include RNA-seq analyses of GSE299230 in the main manuscript, with clear QC, normalization, and differential expression methods. Highlight the expression and behavior of DDIT3, GADD45A, THBS2, CCL2, and CSF1R across contrasts, and discuss how these results support (or challenge) your “core” designation.

3. If relevant RNA-seq datasets are scarce, say so plainly in the main text (not only in the supplement) and explain how this limitation affects generalizability.

4. Only 2–3 citations from 2023–2024 are included. Please update the Introduction and Discussion to reflect recent (2023–2025) work that is directly relevant to your question, and explicitly situate your findings relative to these studies.

5. Volcano plots: label the top genes (by |log2FC| and/or smallest FDR) and ensure key candidates (DDIT3, GADD45A, THBS2, CSF1R) are annotated even if they are not in the top N.

6. Provide analysis code, software versions, and complete parameter settings.

7. Limitations paragraph: explicitly address dataset scarcity, potential confounders, and the scope of inference.

Sincerely,

Vinay

---

## [Author Response · Author response to Decision Letter 2]

8 Oct 2025

Response to Reviewers

We appreciate the insightful comments of the Additional Editor. We provided point-to-point responses, and we have revised the manuscript accordingly.

# The current analyses do not convincingly establish the role of the proposed core genes. Please add rigorous computational experiments.

Ans: To strengthen the evidence for our proposed core genes, we performed an independent validation using the RNA-seq dataset GSE299230. RNA-seq validation description with a fully detailed workflow including data acquisition, normalization, and differential expression analysis. These analyses allow us to independently confirm the expression patterns of the proposed core genes and support their relevance in the studied condition. Sections describing this validation have been highlighted in the revised manuscript.

# Please include RNA-seq analyses of GSE299230 in the main manuscript, with clear QC, normalization, and differential expression methods. Highlight the expression and behavior of DDIT3, GADD45A, THBS2, CCL2, and CSF1R across contrasts, and discuss how these results support (or challenge) your “core” designation.

Ans: We have incorporated and highlighted the RNA-seq analyses of GSE299230 into the main manuscript.

# If relevant RNA-seq datasets are scarce, say so plainly in the main text (not only in the supplement) and explain how this limitation affects generalizability.

Ans: We have added a statement in the main text acknowledging that relevant RNA-seq datasets for this condition are limited, which may restrict the generalizability of our findings. We discuss that validation across multiple cohorts remains necessary to confirm the core gene signatures' universality fully.

# Only 2–3 citations from 2023–2024 are included. Please update the Introduction and Discussion to reflect recent (2023–2025) work that is directly relevant to your question, and explicitly situate your findings relative to these studies.

Ans: The Introduction and Discussion sections have been updated to include the most recent studies published between 2023 and 2025 (References 4, 19, 20, 21, 22, 23, 30, 31, 39, 66, 73–78).

# Volcano plots: label the top genes (by |log2FC| and/or smallest FDR) and ensure key candidates (DDIT3, GADD45A, THBS2, CSF1R) are annotated even if they are not in the top N.

Ans: Volcano plots have been updated per the suggestion and are mentioned as Fig. 5D and Fig. 5E in the revised manuscript.

# Provide analysis code, software versions, and complete parameter settings.

Ans: The analysis code can be accessed from the git hub link: https://github.com/Somorita/-Microarray-analysis-for-AGE-responsive-core-genes-in-Diabetic-Nephropathy/tree/main. Software versions and parameter settings are mentioned in the manuscript.

# Limitations paragraph: explicitly address dataset scarcity, potential confounders, and the scope of inference.

Ans: We have added a Limitations paragraph in the Discussion highlighting: Dataset scarcity, Potential confounders, and Scope of inference.

---

## [Editor Report · Decision Letter 2]

15 Oct 2025

Gene Regulatory Networks Involved in Activation of Notch Signaling by AGEs in the Pathogenesis of Diabetic Kidney Disease

PONE-D-25-26105R2

Dear Dr. Pasupulati,

We’re pleased to inform you that your manuscript has been judged scientifically suitable for publication and will be formally accepted for publication once it meets all outstanding technical requirements.

Kind regards,

Vinay Randhawa, Ph.D.

Academic Editor

PLOS ONE

---

## [Editor Report · Acceptance letter]

PONE-D-25-26105R2

PLOS One

Dear Dr. Pasupulati,

I'm pleased to inform you that your manuscript has been deemed suitable for publication in PLOS One. Congratulations! Your manuscript is now being handed over to our production team.

Kind regards,

on behalf of

Dr. Vinay Randhawa

Academic Editor

PLOS One